# Bayesian Risk-Averse Q-Learning with Streaming Observations

**Yuhao Wang**
School of Industrial and Systems Engineering
Georgia Institute of Technology
Atlanta, GA 30332
yuhaowang@gatech.edu

**Enlu Zhou**
School of Industrial and Systems Engineering
Georgia Institute of Technology
Atlanta, GA 30332
enlu.zhou@isye.gatech.edu

## Abstract

We consider a robust reinforcement learning problem, where a learning agent learns from a simulated training environment. To account for the model misspecification between this training environment and the real environment due to lack of data, we adopt a formulation of Bayesian risk MDP (BRMDP) with infinite horizon, which uses Bayesian posterior to estimate the transition model and impose a risk measure to account for the model uncertainty. Observations from the real environment that are out of the agent's control arrive periodically and are utilized by the agent to update the Bayesian posterior to reduce model uncertainty. We theoretically demonstrate that BRMDP balances the trade-off between robustness and conservativeness, and we further develop a multi-stage Bayesian risk-averse Q-learning algorithm to solve BRMDP with streaming observations from real environment. The proposed algorithm learns a risk-averse yet optimal policy that depends on the availability of real-world observations. We provide a theoretical guarantee of strong convergence for the proposed algorithm.

## 1   Introduction

Markov Decision Process (MDP) is widely applied in the Reinforcement Learning (RL) community to model sequential decision-making, where the underlying transition process is a Markov Process for a given policy. The Q-learning algorithm, which was first proposed in [28], learns the optimal Q-function of the infinite-horizon MDP. The Q-function is a function in terms of the state and action pair $(s, a)$, which denotes the expected reward when action $a$ is taken at the initial state $s$ and optimal action is taken at subsequent stages. In a Q-learning algorithm, the decision maker updates the value of the Q-function by constantly interacting with the training environment and observing the reward and transition each time. The training environment is often the estimate of the real environment (where the policy is actually implemented) through past observed data. Implementing the policy learned from the training environment in the real environment directly can be risky, because of model misspecification, which is caused by the lack of historical data and possible environment shift. For example, in an inventory management problem where the demand follows some unknown distribution, we have limited observations of past demand realizations or only partially observed data if unfulfilled lost orders are not observed.

One common approach is to consider a distributionally robust formulation, where one assumes the embedding unknown distribution belongs to some ambiguity set. This ambiguity set is usually chosen as a neighborhood of the reference distribution (estimated from data). For example, [23] considered the distributionally robust policy learning for the contextual bandit problems, and [32, 17, 30] studied the distributionally robust policy learning for the discounted RL setting. In particular, [14] and [15] considered Q-learning methods for distributionally robust RL with a discounted reward, where [14] constructed the ambiguity set using the Kullback-Leibler (KL) divergence whereas [15] used the

37th Conference on Neural Information Processing Systems (NeurIPS 2023).

Wasserstein distance; Both works designed the Q-learning algorithm by reformulating the robust Bellman equation in its dual form, and thus, transforming the optimization over the distributions to the optimization over a scalar.

While distributionally robust formulation accounts for model misspecification and provides a robust policy that has relatively good performance over all possible distributions in the ambiguity set, it is also argued to be conservative as a price for robustness, especially when the worst-case scenario is unlikely to happen in reality. This motivates us to take a more flexible risk quantification criterion instead of only considering the worst-case performance. [31, 29] proposed a Bayesian risk optimization (BRO) framework, which adopts the Bayesian posterior distribution, as opposed to an ambiguity set, to quantify distributional parameter uncertainty. With this quantification, a risk measure, which represents the user's attitude towards risk, is imposed on the objective with respect to the posterior distribution. In this paper, we take this Bayesian risk perspective and formulate the infinite-horizon Bayesian risk MDP (BRMDP) with discounted reward.

BRMDP was first proposed in [13], where they model the unknown parameter of the MDP via a Bayesian posterior distribution. The posterior distribution is absorbed as an augmented state, together with the original physical state, and is updated at each stage through the realization of the reward and transition at that stage. A risk measure is imposed on the future reward at each stage taken with respect to the posterior distribution at that stage. Before [13], using the Bayesian technique in MDP was considered in [8] to deal with parameter uncertainty, where they simply took expectation on the total reward with respect to the posterior distribution and referred to this MDP model as Bayes-adaptive MDP (BAMDP). Based on BAMDP, there is a stream of work on model-based Bayesian RL (e.g., [25, 7, 27, 18, 3, 20, 4, 16]). Other model-free Bayesian RL methods include Gaussian process temporal difference learning [9], Bayesian policy gradient methods [11], and Bayesian actor-critic methods [26]. We referred to [12] for a comprehensive overview of Bayesian RL. More recently, [22] and [19] considered risk-sensitive BAMDP by imposing the risk measure on the total reward. In addition, [5] considered a percentile criterion and formulated a second-order cone programming by assuming a Gaussian random reward. All three works mentioned above took a non-nested formulation i.e., only one risk measure is imposed on the total reward. As pointed out in [13], one benefit of the nested formulation (i.e., a risk measure is imposed at each stage) is that the resulting optimal policy is time consistent, meaning the policy solved at the initial stage remains optimal at any later stage. In contrast, the non-nested formulation does not have this property (see [21]). In this paper, we considered the nested formulation as in [13]. Notably, [13] as well as most works of BAMDP considered the planning problem over a finite horizon. The problem can be solved efficiently only when the number of horizons is small since the number of augmented states, i.e., possible posterior distributions, increases exponentially as the time horizon increases. In this paper, we formulate the infinite-horizon BRMDP whose state space only contains the physical states. As a result, this enables us to find the stationary (time-invariant) optimal policy. We develop a Q-learning algorithm to learn this stationary optimal policy, which is quite different from the dynamic programming approach taken by [13].

On a related note, risk measures are also widely applied in the literature on safe RL (see [10] for a recent survey). In safe RL, risk measures are used to ensure that the learned policy not only performs well but also avoids dangerous states or actions that can cause large one-stage costs. risk measures are typically applied to deal with the intrinsic uncertainty of the MDP, taking into account the known transition probability or reward distribution. In contrast, our work uses risk measures to account for parametric uncertainty in the transition probability, which can prevent finding the optimal policy. Moreover, in safe RL, the risk measure is usually imposed on some cost function to satisfy some safety constraint, while we impose it on the reward objective. Although both areas use risk measures, they have different goals and frameworks.

To summarize, all the pre-mentioned work on robust RL has focused on offline learning where the agent does not interact with the real environment, but only has access to data from the training environment. In contrast, our work utilizes real-world data to update the Bayesian posterior in the offline learning process to reduce the parametric uncertainty. We also differ from Bayesian online learning, where the Bayesian posterior is updated periodically, and another optimal policy is then re-solved (regardless of computational cost) and deployed. Our approach is off-policy learning, where the learning agent receives streaming real-world observations from some behavior policy, and we focus on designing a Q-learning algorithm for the varying BRMDP that possesses statistical validity rather than simply assuming optimality is obtained after each posterior update.

## 2 Preliminary and Problem Statement

### 2.1 Standard Reinforcement Learning

Consider an RL environment $\mathcal{M}^c = (\mathcal{S}, \mathcal{A}, \mathcal{P}^c, r, \gamma)$, where $\mathcal{S}$ is a finite state space, $\mathcal{A}$ is a finite action space, $r : \mathcal{S} \times \mathcal{A} \times \mathcal{S} \to \mathbb{R}$ is the reward function bounded by $\bar{R} = \max_{s,a,s'} |r(s,a,s')|$, $\mathcal{P}^c = \{p^c_{s,a}(\cdot)\}_{(s,a) \in \mathcal{S} \times \mathcal{A}}$ is the transition probabilities, and $\gamma$ is the discount factor. The standard reinforcement learning aims to find a (deterministic) optimal policy $\pi : \mathcal{S} \to \mathcal{A}$ satisfying

$$V^\pi(s) = \mathbb{E}_{s' \sim p^c_{s,\pi(s)}}[r(s, \pi(s), s') + \gamma V^\pi(s')] = \sup_{a \in \mathcal{A}} \left\{ \mathbb{E}_{s' \sim p^c_{s,a}}[r(s, a, s') + \gamma V^\pi(s')] \right\}.$$

### 2.2 Bayesian Risk Markov Decision Process (BRMDP)

The true transition probabilities, $\mathcal{P}^c$, are usually unknown in real problems, and we need to estimate them using observations from the real world. However, as discussed before, model misspecification due to lack of data can impair the performance of learned policy when deployed in the real environment. Hence, we take a Bayesian approach to estimate the environment and impose a risk measure on the objective with respect to the Bayesian posterior to account for this model (parametric) uncertainty. With finite state and action spaces, we can impose a Dirichlet conjugate prior $\phi_{s,a}$ on the transition model of each state-action $(s, a)$ pair and update the Bayesian posterior once we observe a transition from state $s$ to $s'$ by taking action $a$. We defer the details of updating the Dirichlet posterior on $(s, a)$ to the supplementary material.

The Bayesian posterior provides a quantification of uncertainty about the transition model, which a risk-sensitive decision maker seeks to address by making robust decisions using the current model estimate. This can be done by imposing a risk measure on the future reward at each stage, which maps a random variable to a real value and reflects different attitudes toward risk. Given a Dirichlet posterior $\phi = \{\phi_{s,a}\}_{s,a}$ and a risk measure $\rho_\xi(f(\xi))$ that maps a function $f$ of the random vector $\xi$ to a real value, the value function of BRMDP under a policy $\pi$ is defined as

$$\begin{aligned} V^{\phi,\pi}(s_0) = &\mathbb{E}_{d_0 \sim \pi(s_0)} \{ \rho_{p_1 \sim \phi_{s_0,d_0}} \left( \mathbb{E}_{s_1 \sim p_1}[r(s_0, d_0, s_1) + \right. \\ &\gamma \mathbb{E}_{d_1 \sim \pi(s_1)} \{ \rho_{p_2 \sim \phi_{s_1,d_1}} \left( \mathbb{E}_{s_2 \sim p_2}[r(s_1, d_1, s_2) + \right. \\ &\gamma \mathbb{E}_{d_2 \sim \pi(s_2)} \{ \rho_{p_3 \sim \phi_{s_2,d_2}} \left( \mathbb{E}_{s_3 \sim p_3}[r(s_2, d_2, s_3) + \cdots \right. \end{aligned} \quad (1)$$

We assume the risk measure $\rho$ satisfies the following assumption.

**Assumption 2.3.** Let $\xi \in \mathbb{P}(\mathbb{R}^n)$ denote a random vector taking values in $\mathbb{R}^n$, and let $f_i : \mathbb{R}^n \to \mathbb{R}$, $i = 1, 2$ denote two measurable functions. The risk measure satisfies the following conditions:

1. $\rho_\xi(\gamma f_1(\xi)) = \gamma \rho(f_1(\xi))$ for all $\gamma \geq 0$.

2. $\rho_\xi(f_1(\xi)) \geq \rho_\xi(f_2(\xi))$ if $f_1(\xi) \geq f_2(\xi)$ almost surely.

3. $\rho_\xi(f_1(\xi) + C) = \rho_\xi(f_1(\xi)) + C$ for all constant $C$.

risk measures satisfying Assumption 2.3 are similar to the coherent risk measures (see [1]), except that the sub-additivity is not required. We relax the assumption of coherent risk measure to include the risk measure VaR, which does not admit the sub-additivity.

Notice (1) takes a nested formulation, where the risk measure is imposed on the future reward at each stage as opposed to the non-nested formulation where the risk measure is only imposed once on the total reward, i.e., the value function of the classic MDP. One advantage of the nested formulation is that (1) can be expressed in a recursive form:

$$V^{\phi,\pi}(s) = \mathbb{E}_{a \sim \pi(s)} \{ \rho_{p \sim \phi_{s,a}} \left( \mathbb{E}_{s' \sim p}[r(s, a, s') + \gamma V^{\phi,\pi}(s')] \right) \}.$$

This enables us to define the Bellman operator to find the time-invariant optimal policy. Let $\pi^*$ denote the optimal policy such that $V^{\phi,*} := V^{\phi,\pi^*}$ satisfying

$$\begin{aligned} V^{\phi,*}(s_0) = \sup_{\pi \in \Pi} \{ &\mathbb{E}_{d_0 \sim \pi(s_0)} \{ \rho_{p_1 \sim \phi_{s_0,d_0}} \left( \mathbb{E}_{s_1 \sim p_1}[r(s_0, d_0, s_1) + \right. \\ &\gamma \mathbb{E}_{d_1 \sim \pi(s_1)} \{ \rho_{p_2 \sim \phi_{s_1,d_1}} \left( \mathbb{E}_{s_2 \sim p_2}[r(s_1, d_1, s_2) + \right. \\ &\gamma \mathbb{E}_{d_2 \sim \pi(s_2)} \{ \rho_{p_3 \sim \phi_{s_2,d_2}} \left( \mathbb{E}_{s_3 \sim p_3}[r(s_2, d_2, s_3) + \cdots \right. \end{aligned}$$

where $\Pi$ contains all randomized (and deterministic) policies. Let $\mathcal{L}^\phi$ be the Bellman operator such that

$$\mathcal{L}^\phi V(s) = \max_{a \in \mathcal{A}} \rho_{p \sim \phi_{s,a}} (\mathbb{E}_{s' \sim p}[r(s, a, s') + \gamma V(s')]).$$

The following theorem ensures that $\mathcal{L}^\phi$ is a contraction mapping and BRMDP admits an optimal value function which is the unique fixed point of $\mathcal{L}^\phi$. Its proof is in the supplementary material.

**Theorem 2.4.** *BRMDPs possess the following properties:*

1. *$\mathcal{L}^\phi$ is a contraction mapping with $||\mathcal{L}^\phi V - \mathcal{L}^\phi U||_\infty \leq \gamma ||V - U||_\infty$, where $|| \cdot ||_\infty$ is the sup norm in $\mathbb{R}^{|\mathcal{S}|}$.*

2. *There exists a unique $V^{\phi,*}$ such that $\mathcal{L}^\phi V^{\phi,*} = V^{\phi,*}$. Moreover,*

$$V^{\phi,*}(s_0) = \sup_{\pi \in \Pi}\{\mathbb{E}_{d_0 \sim \pi(s_0)}[\rho_{p_1 \sim \phi_{s_0, d_0}} (\mathbb{E}_{s_1 \sim p_1}[r(s_0, d_0, s_1)+$$

$$\gamma \mathbb{E}_{d_1 \sim \pi(s_1)}[\rho_{p_2 \sim \phi_{s_1, d_1}} (\mathbb{E}_{s_2 \sim p_2}[r(s_1, d_1, s_2)+$$

$$\gamma \mathbb{E}_{d_2 \sim \pi(s_2)}[\rho_{p_3 \sim \phi_{s_2, d_2}} (\mathbb{E}_{s_3 \sim p_3}[r(s_2, d_2, s_3) + \ldots$$

## 2.5   BRMDP with VaR and CVaR

Among choices of risk measures that satisfy Assumption 2.3, we adopt two of the most commonly used risk measures, VaR and CVaR. For a random variable $X$ properly defined on some probability space, $\text{VaR}_\alpha$ is the $\alpha$-quantile of $X$, $\text{VaR}_\alpha(X) = \inf\{z | \mathbb{P}(X \leq z) \geq \alpha\}$, and $\text{CVaR}_\alpha$ normally is defined as the mean of $\alpha$-tail distribution of $X$. However, since we are imposing the risk measure on the reward (regarded as the negative of loss) rather than the loss, we define $\text{CVaR}_\alpha(X) = \frac{1}{\alpha} \int_{-\infty}^{\text{VaR}_\alpha(X)} x \mathbb{P}(\mathrm{d}x)$, which is the conditional expectation for $X \leq \text{VaR}_\alpha$.

Before discussing solving the BRMDP, a natural question is how this risk-averse formulation is related to the original risk-neutral objective. Let $V^{c,\pi}$ denote the value function of policy $\pi$ under the true MDP $\mathcal{M}^c$. the difference between $V^{\phi,\pi}$ and $V^{c,\pi}$ is bounded by the following theorem.

**Theorem 2.6.** *Suppose the Bayesian prior $\phi_{s,a}^0(s') = 1, \forall s, a, s'$ and $\rho$ is either $\text{VaR}_\alpha$ or $\text{CVaR}_\alpha$. Let $\pi$ be a deterministic policy. Let $\bar{O} = \min_{s \in \mathcal{S}} O_{s, \pi(s)}$ and $O_{s,a}$ be the number of observed transitions from $s$ with action $\pi(s)$ in the dataset. Assuming $O_{s,a} > 0$ for all $(s, a)$, then with probability at least $1 - \bar{O}^{-\frac{1}{3}}\sqrt{\frac{|\mathcal{S}|}{\alpha}}$,*

$$\|V^{\phi,\pi} - V^{c,\pi}\|_\infty \leq \bar{O}^{-\frac{1}{3}}\sqrt{\frac{|\mathcal{S}|}{\alpha}} \frac{5|\mathcal{S}|\bar{R}}{(1-\gamma)^2},$$

*In particular, let $\underline{O} = \min_{s \in \mathcal{S}, a \in \mathcal{A}} O_{s,a}$, then with probability at least $1 - \underline{O}^{-\frac{1}{3}}\sqrt{\frac{|\mathcal{S}|}{\alpha}}$,*

$$\|V^{\phi,*} - V^{c,*}\|_\infty \leq \underline{O}^{-\frac{1}{3}}\sqrt{\frac{|\mathcal{S}|}{\alpha}} \frac{5|\mathcal{S}|\bar{R}}{(1-\gamma)^2}$$

*where $V^{c,*}$ is the optimal value function for $\mathcal{M}^c$.*

Theorem 2.6 implies our reformulated BRMDP coincides with the risk-neutral MDP as more observations are available. In particular, the discrepancy is characterized in terms of the minimal number of observations for any state-action pair with the order at least $\underline{O}^{-\frac{1}{3}}$. This indicates the BRMDP automatically balances the trade-off between robustness and conservativeness with varying numbers of observations. We defer the proof to the supplementary material due to the space limit.

## 2.7   Q-Learning for BRMDP

Section 2.2 ensures BRMDP is well-formulated, that is, it admits an optimal value function which can be found as the fixed point of its Bellman operator $\mathcal{L}^\phi$. This allows us to derive a Q-Learning algorithm to learn the optimal policy and value function of BRMDP. Recall that an optimal Q-function is defined as

$$Q^{\phi,*}(s, a) = \rho_{p \sim \phi_{s,a}} (\mathbb{E}_{s' \sim p}[r(s, a, s') + \gamma V^{\phi,*}(s')]) = \rho_{p \sim \phi_{s,a}} (\mathbb{E}_{s' \sim p}[r(s, a, s') + \gamma \max_{b \in \mathcal{A}} Q^{\phi,*}(s', b)]).$$

Let $\mathcal{T}^\phi$ denote the Bellman operator for the Q-function. That is,

$$\mathcal{T}^\phi Q(s,a) = \rho_{p \sim \phi_{s,a}}(\mathbb{E}_{s' \sim p}[r(s,a,s') + \gamma \max_{b \in \mathcal{A}} Q(s',b)]).$$

The optimal Q-function $Q^{\phi,*}(s,a)$ then satisfies

$$Q^{\phi,*}(s,a) = (1-\lambda)Q^{\phi,*}(s,a) + \lambda \mathcal{T}^\phi Q^{\phi,*}(s,a),$$

where $\lambda \in (0,1)$. In a Q-learning algorithm, given a sequence of learning rates $\{\lambda_t\}_t$ and an estimator of the Bellman operator $\widehat{\mathcal{T}}^\phi$, we have the Q-learning update rule

$$Q_{t+1}(s,a) = (1-\lambda_t)Q_t(s,a) + \lambda_t \widehat{\mathcal{T}}^\phi Q_t(s,a).$$

## 2.8 Bayesian Risk-Averse Q-Learning with Streaming Data

Notice that in Section 2.2 and 2.7, the posterior distribution $\phi$ is fixed across stages. As in many previous works, the embedding model is estimated with a fixed set of past observations before solving the problem. However, this one-time estimation of the transition model does not take into consideration utilizing the new data that arrive later, which helps reduce the model uncertainty as well as the conservativeness caused by risk measure, as indicated by Theorem 2.6. This motivates us to consider a data-driven framework where we dynamically update the estimate of the model. For this purpose, we consider a multi-stage Bayesian Q-learning algorithm.

Suppose at the beginning of stage $t$, a batch of observations in the form of three-tuple $(s_i, a_i, s_i')$ with batch size $n(t)$ is available. The observation can be regarded as generated by some policy that is actually deployed in the real world, which does not depend on the learning process. The decision maker incorporates these new data to update the posterior $\phi^t$, with which we obtain a BRMDP model $\mathcal{M}_t$. We then cary out $m(t)$ steps of the Q-learning update, where the initial Q-function in stage $t$ is inherited from the previous stage $t-1$. This framework is shown in Figure 1.

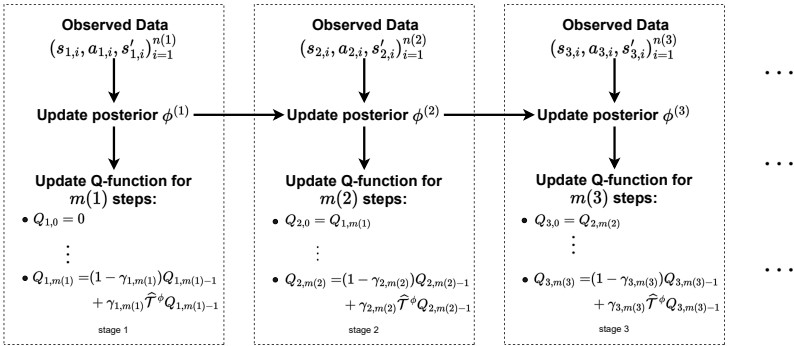

Figure 1: Multi-stage Bayesian risk-averse Q-learning

## 3 Estimator for Bellman Operator

In Figure 1, a key step is to design a proper estimator $\widehat{\mathcal{T}}^\phi$ for the Bellman operator to ensure the convergence of the Q-function. In most of the existing literature on Q-learning, convergence relies on an unbiased estimator of the Bellman operator. While this is usually easy to obtain with the expectation operator which is linear, in BRMDP unbiased estimator for the Bellman operator is difficult if not impossible to obtain, because of (i) the non-linearity of risk measure (VaR and CVaR) and (ii) varying posterior distributions. Unbiased estimators for nonlinear functionals have been studied in [2]; however, their method cannot be directly applied here since the variance of the estimator is uncontrollable in the existence of varying posteriors. Instead, we use the Monto Carlo simulation to obtain an estimator with almost surely diminishing bias. We show in Theorem 4.3 and 4.4 that the Monto Carlo estimator is sufficient to guarantee the convergence of the Q-function.

## 3.1 Monto Carlo Estimator for VaR and CVaR with Varying Sample Size

Denote by

$$f(p|s, a, Q) = \mathbb{E}_{s' \sim p}[r(s, a, s') + \gamma \max_{b \in \mathcal{A}} Q(s', b)] = \sum_{s' \in \mathcal{S}} p(s')[r(s, a, s') + \gamma \max_{b \in \mathcal{A}} Q(s', b)]. \quad (2)$$

Given a posterior $\phi$ and a Q-function $Q$, we want to estimate

$$\mathcal{T}^\phi Q(s, a) = \rho_{p \sim \phi_{s,a}}(f(p|s, a, Q)),$$

where $\rho$ is either $\text{VaR}_\alpha$ or $\text{CVaR}_\alpha$ and $\alpha \in (0, 1)$ denotes the risk level. A Monto Carlo estimator for $\text{VaR}_\alpha(f(p|s, a, Q))$ and $\text{CVaR}_\alpha(f(p|s, a, Q))$ with sample size $N$ can be obtained by first drawing independent and identically distributed (i.i.d.) samples $p_1, p_2, \ldots, p_N \sim \phi_{s,a}$. Denote by $X_i := f(p_i|s, a, Q)$, $i = 1, 2, \ldots, N$. Then

$$\widehat{\mathcal{T}}_N^\phi Q(s, a) = \begin{cases} \widehat{\text{VaR}}_{\alpha,N}^{\phi_{s,a}}(f(p|s, a, Q)) := X_{\lceil N\alpha \rceil:N} & \text{if } \rho \text{ is } \text{VaR}_\alpha, \\ \widehat{\text{CVaR}}_{\alpha,N}^{\phi_{s,a}}(f(p|s, a, Q)) := \frac{1}{\lceil N\alpha \rceil} \sum_{k=1}^{\lceil N\alpha \rceil} X_{k:N} & \text{if } \rho \text{ is } \text{CVaR}_\alpha, \end{cases} \quad (3)$$

where $X_{k:N}$ is the $k$th order statistic out of $N$ samples. Both estimators are biased with a constant sample size. Increasing the sample size reduces the bias but can be computationally inefficient. As the posterior distribution concentrates more on the true model, even estimators with small sample sizes can have reduced bias. We use a varying sample size that adapts to the updated posterior distribution. Initially, we set $N_{s,a}$ as a fixed minimal sample size $N$, and decrease or increase $N_{s,a}$ by 1 at the beginning of each stage based on whether a new observation improves or not the estimate for the transition model on $(s, a)$. The multi-stage Bayesian Risk-averse Q-Learning algorithms with VaR (BRQL-VaR) and CVaR (BRQL-CVaR) are presented in Algorithm 1.

## 4 Convergence Analysis

In this section, we give the theoretical guarantee of our proposed multi-stage Bayesian risk-averse Q-learning algorithm, which ensures the asymptotic convergence of the learned Q-function to the "optimal" Q-function. We define the random observations from the real environment and all algorithm-related random quantities with respect to a probability space $(\Omega, \mathcal{F}, \mathbb{P})$.

Recall that the real-world transition observations are assumed to be obtained from some behavior policy that is not controlled by the agents. This may possibly bring a definition problem of "optimal" Q-function. Indeed, if all streaming data are generated by some greedy policy, then the number of transition observations from some state-action pair can remain small, in which case the Bayesian posterior does not converge and the learned policy is risk-averse yet random since it depends on the Bayesian posterior. As in the offline RL, where the policy is learned based on the initial fixed data set, we also define the optimal Q-function as depending on the given real-world data but differs from pure offline RL in that the given data is streaming and random.

**Definition 4.1. (Data-conditional optimal Q-function)**
Given an observation process $\omega = \{(s_{t,i}, a_{t,i}, s'_{t,i})|t = 1, 2, \ldots, i = 1, 2, \ldots, n(t)\}$, let $O_{s,a,s'}^\infty(\omega) = \sum_{t=1}^\infty \sum_{i=1}^{n(t)} \mathbf{1}\{(s_{t,i}, a_{t,i}, s'_{t,i}) = (s, a, s')\}$ and $O_{s,a}^\infty(\omega) = (O_{s,a,s'}^\infty(\omega))_{s' \in \mathcal{S}}$ denote the number of transition observations under $\omega$. For each $(s, a) \in \mathcal{S} \times \mathcal{A}$, define the limiting posterior as

$$\phi_{s,a}^\omega = \begin{cases} \delta_{p_{s,a}^c}(\cdot) & \text{if } ||O_{s,a}^\infty(\omega)|| = \infty, \\ \text{Dirichlet}\left(\phi_{s,a}^0 + O_{s,a}^\infty(\omega)\right) & \text{otherwise,} \end{cases}$$

where $\delta_{p_{s,a}^c}(\cdot)$ is the Dirac measure centered at true transition probability $p_{s,a}^c = (p_{s,a}^c(s'))_{s' \in \mathcal{S}}$. $Q^{\omega,*}$ is the optimal solution to BRMDP with "posterior" $\phi^\omega$, which satisfies $Q^{\omega,*} = \mathcal{T}^{\phi^\omega} Q^{\omega,*}$. The data-optimal Q-function $Q^{\omega,*}$ is a random vector since the observation process is random.

With this data-conditional optimal criterion, we can now prove the convergence of the multi-stage Bayesian risk-averse Q-learning algorithm. A list of notations for different Q-functions can be found in Table 1 of the supplementary material.

---
**Algorithm 1** Multi-stage Bayesian risk-averse Q-learning
---
**Input:** State space $\mathcal{S}$, action space $\mathcal{A}$, reward function $r$, termination stage $T$, Q-learning update step size $\{m(t)\}_{t=1}^{T}$, learning rate $\{\lambda_\ell\}_\ell$, prior distribution $\{\phi_{s,a}^0\}_{s\in\mathcal{S},a\in\mathcal{A}}$, minimal sample size $N$, risk measure $\rho \in \{\text{VaR}_\alpha, \text{CVaR}_\alpha\}$ with risk level $\alpha$.
**Initialize** $\phi \leftarrow \phi^0$, $Q(s,a) \leftarrow 0$, $N_{s,a} \leftarrow N \ \forall s,a$ .
**for** $t = 1$ **to** $T$ **do**
    Obtain $n(t)$ observations $(s_i, a_i, s_i')$ $i = 1, \ldots, n(t)$.
    **for** $i = 1$ **to** $n(t)$ **do**
        **for all** $(s,a) \in \mathcal{S} \times \mathcal{A}$ **do**
            $\phi_{s,a}^t(s') \leftarrow \phi_{s,a}^{t-1}(s') + \#\{(s_i, a_i, s_i') = (s, a, s')\}$
            **if** $\phi_{s,a}^t = \phi_{s,a}^{t-1}$ **then**
                $N_{s,a} \leftarrow N_{s,a} + 1$
            **else**
                $N_{s,a} \leftarrow \max\{N_{s,a} - 1, N\}$
            **end if**
        **end for**
    **end for**
    **for** $\ell = 1$ **to** $m(t)$ **do**
        $M \leftarrow \sum_{\tau=1}^{t-1} m(t)$, $\lambda_{t,l} \leftarrow \lambda_{M+\ell}$
        **for all** $(s,a) \in \mathcal{S} \times \mathcal{A}$ **do**
            Generate $p_1, \ldots, p_{N_{s,a}} \sim \phi_{s,a}^t$
            $X_i \leftarrow f(p_i|s, a, Q), i = 1, 2, \ldots, N_{s,a}$
            $Q'(s,a) \leftarrow (1 - \lambda_{t,\ell})Q(s,a) + \lambda_{t,\ell}\widehat{T}_{N_{s,a}}^{\phi^t} Q(s,a)$ using (3).
        **end for**
        $Q(s,a) \leftarrow Q'(s,a) \ \forall (s,a) \in \mathcal{S} \times \mathcal{A}$
    **end for**
**end for**
**Output: Q-function** $\{Q(s,a)\}_{(s,a)}$
---

To prove the convergence of $Q_t$, we prove (i) the convergence of $Q^{\phi^t,*}$ which depends on the convergence of the posterior distribution, and (ii) the convergence of the estimator for the Bellman operator. We summarize two important results in Proposition 4.2 and Theorem 4.3. Due to the page limit, all the proofs are deferred to the supplementary material.

**Proposition 4.2.** *(Convergence of $Q^{\phi^t,*}$) Let $Q^{\omega,*}$ be defined as in Definition 4.1 and $\phi^t$ be the posterior distribution obtained at stage $t$. Then the optimal Q-function under posterior distribution $\phi^t$ converges to $Q^{\omega,*}$ almost surely. That is,*

$$\lim_{t\to\infty} ||Q^{\phi^t,*} - Q^{\omega,*}||_\infty = 0 \qquad almost\ surely,$$

*where $|| \cdot ||_\infty$ is the entry-wise sup norm.*

Compared with Theorem 2.6 which characterizes the difference between the BRMDP and the original MDP by computing a concentration bound, Proposition 4.2 ensures the strong convergence of the BRMDP model to the data-conditional optimal Q-function. The next Theorem 4.3 ensures that the bias of the proposed estimator for the Bellman operator with varying sample sizes converges to zero uniformly in $Q$.

**Theorem 4.3.** *(Diminishing bias for Bellman estimator) Denote by $(k)$ the kth iteration of the Q-learning update. Let $t_k$ be the stage such that $\sum_{t=1}^{t_k-1} m(t) < k \le \sum_{t=1}^{t_k} m(t)$. Let $N_{s,a}^{(k)}$ denote the sample size in iteration $k$ and $\omega_{t_k}$ denote all the past observations until stage $t_k$. Denote by $\widehat{\mathcal{T}}^{(k)}Q(s,a) = \widehat{\mathcal{T}}_{N_{s,a}^{(k)}}^{\phi_{t_k}} Q(s,a)$ as defined in (3). Then, we have almost surely,*

$$\lim_{k\to\infty} \sup_{||Q||_\infty \le \frac{\bar{R}}{1-\gamma}} \left| \mathbb{E}[\widehat{\mathcal{T}}^{(k)}Q(s,a) - \mathcal{T}^{\phi_{t_k}}Q(s,a)|\omega_{t_k}] \right| = 0.$$

Theorem 4.3 provides a uniform convergence of the estimated Bellman estimator, which is crucial to prove Theorem 4.4. Notably, with both streaming data and risk measure, proof of Theorem 4.3 is

much more challenging than existing work on distributionally robust Q-learning, where randomness only comes from the learning process, which refers to the random data generated by the simulator. In our setting, we have mixed randomness coming from both the observation process and learning process, which complicates the analysis and differs from that of pure offline Q-learning. Also, the analysis is different from online Bayesian Q-learning, where the sample complexity results are usually constructed assuming an optimal policy is resolved and deployed instantly in each period. In contrast, we aim to prove the convergence of Q-learning in order to obtain the optimal policy.

Together with Proposition 4.2 and Theorem 4.3, we establish the convergence of Algorithm 1 in Theorem 4.4.

**Theorem 4.4.** *Denote by $Q_t$ the Q-function given by Algorithm 1 at the end of stage $t$. Assume $T = \infty$, and the learning rate $\{\lambda_\ell\}_{\ell=1}^\infty$ satisfies $\sum_{\ell=1}^\infty \lambda_\ell = \infty$, $\sum_{\ell=1}^\infty \lambda_\ell^2 < \infty$. Then, we have almost surely,*

$$\lim_{t \to \infty} ||Q_t - Q^{\omega,*}||_\infty = 0.$$

**Corollary 4.5.** *Let $Q^t$ be defined as in Theorem 4.4 and $O_{s,a}^\infty$ be defined as in Definition 4.1. Assume $T = \infty$ and $||O_{s,a}^\infty||_\infty = \infty, \forall s \in \mathcal{S}, a \in \mathcal{A}$ almost surely. Then*

$$\lim_{t \to \infty} ||Q^t - Q^{c,*}||_\infty = 0 \quad \text{almost surely} ,$$

*where*

$$Q^{c,*}(s,a) = \sum_{s' \in \mathcal{S}} p_{s,a}^c(s')[r(s,a,s') + \max_{b \in \mathcal{A}} Q^{c,*}(s',b)]$$

*is the true optimal Q-function.*

Corollary 4.5 is straightforward from Theorem 4.4, which ensures Algorithm 1 will eventually learn the true optimal Q-function if we can obtain an infinite number of observations for each state-action pair.

## 5 Numerical Experiments

### 5.1 Comparison Baselines:

- BRQL-VaR: our proposed multi-stage Bayesian risk-averse Q-learning algorithm with risk measure VaR;
- BRQL-CVaR: our proposed multi-stage Bayesian risk-averse Q-learning algorithm with risk measure CvaR;
- BRQL-mean: the risk-neutral Bayesian Q-learning function. That is, the risk measure is the expectation taken with respect to the posterior distribution;
- DRQL-KL: distributionally robust Q-learning algorithm with KL divergence (see [14]);
- DRQL-Wass: distributionally robust Q-learning algorithm with Wasserstein distance (see [15]).

### 5.2 Testing examples

**Example 1: Coin Toss.** Consider we are playing the following game. Each time we will toss K coins and observe the number of coins that show heads, where the chance of each coin showing heads is unknown. After observing the number of heads in the last toss, we can make a guess about whether the next toss will have more heads or fewer heads. If our guess is right, we can get 1 dollar as the reward, otherwise, we need to pay 1 dollar. We also have the choice of not guessing, in which case we do not pay or get paid. We model this game as a discounted infinite-horizon MDP. The state $s_t \in \mathcal{S} = \{0, 1, \ldots, K\}$ denotes the number of heads in $t$th toss. The actions space is $\mathcal{A} = \{-1, 0, 1\}$, where $a = 1$ corresponds to guess $s_{t+1} > s_t$, $a = -1$ corresponds to guess $s_{t+1} < s_t$, and $a = 0$ corresponds to not guess. Hence, the reward function is

$$r(s,a,s') := a\mathbf{1}_{\{s<s'\}} - a\mathbf{1}_{\{s>s'\}} - |a|\mathbf{1}_{\{s=s'\}}.$$

Assume each coin shows a head with probability $0.4$. The number of coins $K = 10$.

**Example 2: Inventory Management.** Suppose a warehouse manager runs a capacity system. At the beginning of period $t$, the manager observes the current inventory level $s_t$ and orders additional goods of the amount $a_t$. An ordering cost of $c = 1$ is incurred for each unit of goods. The demand follows a truncated Poisson distribution with a mean of 3 and support $\{0, 1, \ldots, K\}$. Suppose the demand arrives during each period and is fulfilled at the end of the period. For each unit of fulfilled demand, we can obtain a profit of $u = 5$. If a unit of demand is not fulfilled at the end of the period, the demand is lost and a penalty cost of $q = 2$ is incurred. If there are any remaining goods at the end of the period, the goods will be taken to the next period with a holding cost $h = 1$ per unit. The warehouse has a maximal capacity of $K$ for the goods. We model this problem as a discounted infinite-horizon MDP with state space $\mathcal{S} = \{-K, \ldots, 0, \ldots, K\}$, where $(s_t)^+ = \max(s_t, 0)$ represents the inventory level at the beginning of period $t$ and $(s_t)^- = -\min(s_t, 0)$ represents the lost demand at the end of period $t - 1$. The action space is $\mathcal{A}_s := \{0, \ldots, K - s^+\}$. The reward is

$$r(s, a, s') = -(c \cdot a + h \cdot (s')^+ + q \cdot (s')^-) + u \cdot (s^+ + a - (s')^+).$$

We consider two settings: (I) the demand in each period uniformly distributes among $\{0, 1, \ldots, K\}$ and (II) the demand depends on the current inventory level $s$. For the second setting, we will consider the case where observations are insufficient to estimate the transition probability for every state-action pair.

## 5.3 Experiment Setting

**Coin toss.** We consider stage-wise streaming observations with batch size $n(t) = 1$ and stage-wise Q-learning with number of steps $m(t) = 1$. The minimal sample size to estimate the Bellman operator $N = 10$. Initial observed data batch size $n(0) = 10$. We set the radius of the KL ball and the Wasserstein ball to be $0.1$. The risk level for VaR and CVaR is set to $0.2$ in Figure 2 and $0.4$ in Figure 3.

**Inventory Management.** In Figure 4, We set $K = 10, T = 60, m(t) = n(t) = 5, n(0) = 20$, and $N = 10$. The radius of the KL and the Wasserstein ball is $0.05$, and the risk level for VaR and CVaR is $0.2$. In Figure 5, we set the capacity $K = 10$ and the size of the historical data set $n(0) = 30$. The policies are deployed in different environments, where the demand follows different truncated Poisson distributions with means ranging from 3 to 5.

## 5.4 Results

In Figure 2- 4, we compare the value functions of different algorithms as the time stage increases. The value function of each algorithm is calculated by deploying the optimal policy in the real environment. Each curve shows the empirical expected performance and the strip around the curve shows the $95\%$ confidence interval. In both examples, our proposed algorithms outperform the two distributionally robust Q-learning algorithms in both expected performance and variation as the time stage increases, since our proposed algorithms dynamically update the posterior to reduce the model uncertainty while two DRQL algorithms learns with fixed ambiguity set. Compared with the risk-neutral algorithm, BRQL-VaR and BRQL-CVaR achieve lower expected value functions but have smaller variations, which shows the robustness of our two proposed algorithms.

Moreover, in Figure 5 We test for the insufficient data setting, where we only have a set of historical data to estimate the transition model at the initial time stage. The value function is calculated as deploying the learned policy in different environments with demand following different Poisson distributions. Figure 5 indicates the two proposed algorithms achieve higher value functions than the risk-neutral algorithm in the more adversarial setting (with Poisson parameter less than 3), showing their robustness. They also obtain lower value functions than two DRQL algorithms in the adversarial setting and higher value functions in other settings, indicating the risk measure is more flexible compared to the worst-case criterion.

## 6 Conclusion and Limitation

In this paper, we propose a novel multi-stage Bayesian risk-averse Q-learning algorithm to learn the optimal policy with streaming data, by reformulating the infinite-horizon MDP with unknown transition model as an infinite-horizon BRMDP. In particular, we consider the two cases of risk

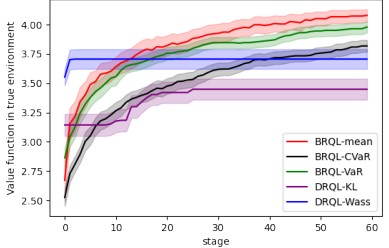

Figure 2: Coin Toss: risk level $\alpha = 0.2$

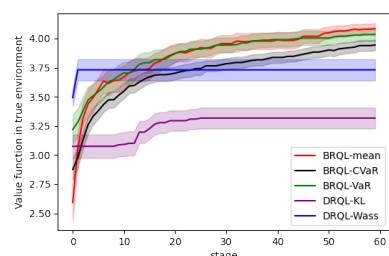

Figure 3: Coin Toss: risk level $\alpha = 0.4$

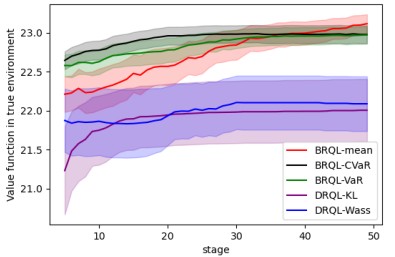

Figure 4: Inventory Management: streaming observations

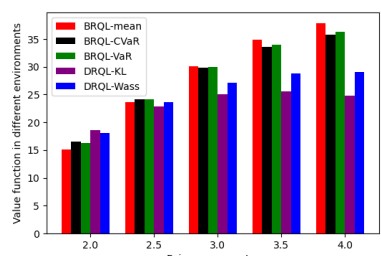

Figure 5: Inventory Management: fixed BR-MDP

measures, VaR and CVaR, for which we design a Monte Carlo estimator with varying sample sizes to approximate the Bellman operator of BRMDP. We demonstrate the correctness of the BRMDP formulation by providing statistical guarantee and prove the strong asymptotic convergence of the proposed Q-Learning algorithm. The numerical results demonstrate that the proposed algorithms are efficient with streaming data and robust with limited data.

As discussed in the paper, one limitation of the current framework is that the behavior policy that generates real-world observations is assumed to be given. This is suitable for situations when it is expensive for the agent to interact with the real environment or to change the policy frequently as it may cause a large cost or system instability. An interesting future direction is to consider an online learning setting, where at each period the agent also needs to take action in the real world.

## ACKNOWLEDGMENT

The authors gratefully acknowledge the support by the Air Force Office of Scientific Research under Grants FA9550-19-1-0283 and FA9550-22-1-0244 and the National Science Foundation under Grant NSF-DMS2053489.

## AUTHOR BIOGRAPHIES

**Yuhao Wang** is a Ph.D. student at the H. Milton Stewart School of Industrial and Systems Engineering at Georgia Institute of Technology. He received his B.S. degree from the Department of Mathematics at Nanjing University, China, in 2021. His research interests include simulation and stochastic optimization and reinforcement learning. His email address is *yuhaowang@gatech.edu* and his web page is `https://sites.gatech.edu/yuhaowang/`.

**Enlu Zhou** is a Professor at the H. Milton Stewart School of Industrial and Systems Engineering at Georgia Institute of Technology. She received the B.S. degree with highest honors in electrical engineering from Zhejiang University, China, in 2004, and received the Ph.D. degree in electrical engineering from the University of Maryland, College Park, in 2009. Her research interests include

simulation optimization, stochastic optimization, and stochastic control. Her email address is *enlu.zhou@isye.gatech.edu*, and her web page is `https://www.enluzhou.gatech.edu/`.

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

## A  Dirichlet Posterior on State-action Pair

A Dirichlet distribution is parameterized by a count vector $\phi = (\phi_1, \ldots, \phi_k)$, where $\phi_i \geq 1$, such that the density of probability distribution $p = (p_1, \ldots, p_k)$ is defined as $f(p \mid \phi) \propto \prod_{i=1}^{k} p_i^{\phi_i - 1}$. For each $(s, a) \in \mathcal{S} \times \mathcal{A}$, we impose a Dirichlet prior with parameter $\phi_{s,a} = (\phi_{s,a}(s'))_{s' \in \mathcal{S}}$ on the unknown transition probability $p_{s,a} = (p_{s,a}(s'))_{s' \in \mathcal{S}}$. After we observe the transition $(s, a, s')$ for $o_{s,a,s'}$ times $\forall s' \in \mathcal{S}$, the posterior distribution of $p_{s,a}$ is also a Dirichlet distribution with parameter $\phi_{s,a} + \mathbf{o}_{s,a} = (\phi_{s,a}(s') + o_{s,a,s'})_{s' \in \mathcal{S}}$, where $\mathbf{o}_{s,a} = (o_{s,a,s'})_{s' \in \mathcal{S}}$.

## B  Technical Proofs

Table 1: Checklist of notations for different Q-functions.

| Notation | Explanation |
|---|---|
| $Q^{\phi,*}$ | Optimal Q-function of BRMDP with posterior $\phi$ |
| $Q^{\omega,*}$ | Data-conditional optimal Q-function conditioned on observation process $\omega$ |
| $Q_t$ | Q-function at stage $t$ given by Algorithm 1 |
| $Q^{c,*}$ | Optimal Q-function of the true environment |

We first introduce the following lemma that guarantees the maximum taken over randomized policies is equivalent to being taken over only deterministic policies.

**Lemma B.1.** *(Deterministic Bellman operator)*

$$\mathcal{L}^\phi V(s) = \max_{a \in \mathcal{A}} \rho_{p \sim \phi_{s,a}}(\mathbb{E}_{s \sim p}[r(s, a, s') + \gamma V(s')])$$
$$= \sup_{\pi \in \Pi} \mathbb{E}_{a \sim \pi(s)}[\rho_{p \sim \phi_{s,a}}(\mathbb{E}_{s' \sim p}[r(s, a, s') + \gamma V(s')])].$$

*Proof.* For an arbitrary randomized policy $\pi$, an arbitrary $V$, denote by $\pi(s, a) = \mathbb{P}(\pi(s) = a)$. We have

$$\sup_{a \in \mathcal{A}} \rho_{p \sim \phi_{s,a}}(\mathbb{E}_{s' \sim p}[r(s, a, s') + \gamma V(s')]) \geq \sum_{a \in \mathcal{A}} \pi(s, a) \rho_{p \sim \phi_{s,a}}(\mathbb{E}_{s' \sim p}[r(s, a, s') + \gamma V(s')]).$$

Hence,

$$\mathcal{L}^\phi V(s) = \max_{a \in \mathcal{A}} \rho_{p \sim \phi_{s,a}}(\mathbb{E}_{s \sim p}[r(s, a, s') + \gamma V(s')]) \geq \sup_{\pi \in \Pi} \mathbb{E}_{a \sim \pi(s)}[\rho_{p \sim \phi_{s,a}}(\mathbb{E}_{s' \sim p}[r(s, a, s') + \gamma V(s')])].$$

The other direction holds naturally since a deterministic policy is also a randomized policy. $\square$

**Lemma B.2.** *(Non-expansive risk measure) Suppose the risk measure $\rho$ satisfies Assumption 2.3, then*
$$|\rho(f_1(\xi)) - \rho(f_2(\xi))| \leq ||f_1(\xi) - f_2(\xi)||_{\xi,\infty},$$
*where $|| \cdot ||_{\xi,\infty}$ is the sup norm with probability measure $\mathbb{P}$.*

*Proof.*

$$\rho_\xi(f_1(\xi)) = \rho_\xi(f_1(\xi) + f_2(\xi) - f_1(\xi)) \leq \rho_\xi(f_1(\xi) + ||f_2(\xi) - f_1(\xi)||_{\xi,\infty}) = \rho_\xi(f_1(\xi)) + ||f_2(\xi) - f_1(\xi)||_{\xi,\infty}.$$

Similarly, we have

$$\rho_\xi(f_2(\xi)) \leq \rho_\xi(f_1(\xi)) + ||f_2(\xi) - f_1(\xi)||_{\xi,\infty}.$$

Combining together we complete the proof. $\square$

**Proof of Theorem 2.4**

*Proof.* **proof of 1.** $\forall \varepsilon > 0$, there exists a deterministic policy $\pi$, such that $\forall s \in \mathcal{S}$

$$\mathcal{L}^\phi V(s) \leq \rho_{p \sim \phi_{s,\pi(s)}}(\mathbb{E}_{s' \sim p}[r(s, \pi(s), s') + \gamma V(s')]) + \varepsilon.$$

Then,

$$\mathcal{L}^\phi V(s) - \mathcal{L}^\phi U(s)$$
$$\leq \rho_{p \sim \phi_{s,\pi(s)}}(\mathbb{E}_{s' \sim p}[r(s, \pi(s), s') + \gamma V(s')]) - \rho_{p \sim \phi_{s,\pi(s)}}(\mathbb{E}_{s' \sim p}[r(s, \pi(s), s') + \gamma U(s')]) + \varepsilon$$
$$\leq \gamma \|\mathbb{E}_{s' \sim p}[V(s') - U(s')]\|_{p,\infty} + \varepsilon$$
$$\leq \gamma \|V - U\|_\infty + \varepsilon$$

Since $\varepsilon$ can be chosen arbitrarily, $\|\mathcal{L}^\phi V - \mathcal{L}^\phi U\| \leq \gamma \|V - U\|_\infty$. Switching the position of $V$ and $U$, we obtain the desired result.

**proof of 2.** For the first part, since $\mathcal{L}^\phi$ is a contraction mapping, there exists a unique $V^{\phi,*}$ such that $\mathcal{L}^\phi V^{\phi,*} = V^{\phi,*}$. For the second part, let $\pi'$ be an arbitrary randomized policy. We have

$$V^{\phi,*}(s) = \mathcal{L}^\phi V^{\phi,*}(s)$$
$$= \sup_{a \in \mathcal{A}} \{\rho_{p_1 \sim \phi_{s,a}}(\mathbb{E}_{s_1 \sim p_1}[r(s, a, s_1) + \gamma V^{\phi,*}(s_1)])\}$$
$$= \sup_{\pi \in \Pi} \sum_{a \in \mathcal{A}} \pi(s, a) \rho_{p_1 \sim \phi_{s,a}}(\mathbb{E}_{s_1 \sim p_1}[r(s, a, s_1) + \gamma V^{\phi,*}(s_1)])$$
$$\geq \sum_{a \in \mathcal{A}} \pi'(s, a) \rho_{p_1 \sim \phi_{s,a}}(\mathbb{E}_{s_1 \sim p_1}[r(s, a, s_1) + \gamma V^{\phi,*}(s_1)])$$

where the third equality holds because of Lemma B.1. Furthermore, by Assumption 2.3.2, we get

$$V^{\phi,*} \geq \sum_{a \in \mathcal{A}} \pi'(s, a) \rho_{p_1 \sim \phi_{s,a}}(\mathbb{E}_{s_1 \sim p_1}[r(s, a, s_1) + \gamma \sum_{a_1 \in \mathcal{A}} \pi'(s_1, a_1) \rho_{p_2 \sim \phi_{s_1,a_1}}(\mathbb{E}_{s_2 \sim p_2}[r(s_1, a_1, s_2) + \gamma V^{\phi,*}(s_2)])]).$$

(4)

Define $\mathcal{L}^\phi_{\pi'}$ such that $\forall s \in \mathcal{S}$ and $V \in \mathbb{R}^{|\mathcal{S}|}$,

$$\mathcal{L}^\phi_{\pi'} V(s) = \sum_{a \in \mathcal{A}} \pi'(s, a) \rho_{p_1 \sim \phi_{s,a}}(\mathbb{E}_{s_1 \sim p_1}[r(s, a, s_1) + \gamma V(s_1)]).$$

By the same induction as (4), we can derive

$$V^{\phi,*} \geq (\mathcal{L}^\phi_{\pi'})^k V^{\phi,*}, \quad \forall k \geq 1.$$

Define

$$V^{0,\pi'}(s) = \sum_{a \in \mathcal{A}} \pi'(s, a) \rho_{p \sim \phi_{s,a}}(\mathbb{E}_{s_1 \sim p}[r(s, a, s_1)])$$

and

$$V^{k+1,\pi'}(s) = \sum_{a \in \mathcal{A}} \pi'(s, a) \rho_{p \sim \phi_{s,a}}(\mathbb{E}_{s_1 \sim p}[r(s, a, s_1) + \gamma V^{k,\pi'}(s_1)]).$$

By Assumption 2.3 and Lemma B.2, we have

$$V^{\phi,*}(s) \geq (\mathcal{L}^\phi_{\pi'})^k V^{\phi,*}(s) \geq V^{k,\pi'}(s) - \gamma^k \|V^{\phi,*}\|_\infty$$

(5)

As $k \to \infty$,

$$|V^{\phi,\pi'}(s) - V^{k,\pi'}(s)| \leq \gamma^{k+1} \|V^{\phi,\pi'}\|_\infty \leq \frac{\gamma^{k+1}}{1 - \gamma} \bar{R} \to 0,$$

where $V^{\phi,\pi'}$ is the value function of BRMDP with posterior $\phi$ and policy $\pi'$. Then from (5) we obtain

$$V^{\phi,*}(s) \geq V^{\phi,\pi'}(s) - \gamma^k \|V^{\phi,*}\|_\infty - \frac{\gamma^{k+1}}{1 - \gamma} \bar{R}, \quad \forall s \in \mathcal{S}.$$

By taking $k \to \infty$, we have $V^{\phi,*}(s) \geq V^{\phi,\pi'}(s)$. Since $\pi'$ is chosen arbitrarily, we get the desired result. $\square$

**Proof of Theorem 2.6**

*Proof.* Denote by

$$f(p|s,a,V) = \mathbb{E}_{s'\sim p}[r(s,a,s') + \gamma V(s')] = \sum_{s'\in\mathcal{S}} p(s')[r(s,a,s') + \gamma V(s')].$$

We first bound $|f(p|s,a,V) - f(p'|s,a,V)|$ in terms of $\|p-p'\|_\infty$ for $\|V\|_\infty \leq \frac{\bar{R}}{1-\gamma}$. To see this,

$$
\begin{aligned}
|f(p|s,a,V) - f(p'|s,a,V)| &= \left| \sum_{x\in\mathcal{S}} [p(x) - p'(x)][r(s,a,x) + \gamma V(x)] \right| \\
&\leq |\mathcal{S}|[\bar{R} + \|V\|_\infty] \|p-p'\|_\infty \\
&\leq \frac{|\mathcal{S}|\bar{R}}{1-\gamma} \|p-p'\|_\infty.
\end{aligned}
\tag{6}
$$

Now consider $p \sim \phi_{s,a}$. Since $\phi^0_{s,a} = \mathbf{1}$, we have the posterior parameter $\phi_{s,a}(s') = o_{s,a,s'} + 1$, where $o(s,a,s')$ is the number of transition from $s$ to $s'$ taken action $a$ in the data set. Let $\widehat{\mathcal{M}}$ denote the empirical MDP generated with the same data which are used to estimate $\phi$, with the empirical transition probability $\widehat{p}_{s,a}(s') = \frac{o_{s,a,s'}}{O_{s,a}}$. Let $p^c_{s,a}$ be the true transition distribution. Since $\{o_{s,a,s'}\}_{s'\in\mathcal{S}}$ follows the multinomial distribution with parameter $O_{s,a}; p^c_{s,a}$, we have

$$\mathbb{P}(|\widehat{p}_{s,a}(s') - p^c_{s,a}(s')| \geq \sqrt{\frac{|\mathcal{S}|}{\alpha}} O_{s,a}^{-\frac{1}{3}}) \leq \frac{\alpha}{|\mathcal{S}|} O_{s,a}^{-\frac{1}{3}}$$

by Chebyshev inequality. Define

$$\bar{p}_{s,a}(s') = \mathbb{E}_{p\sim\phi_{s,a}}[p(s')] = \frac{\phi_{s,a}(s')}{\sum_{s''\in\mathcal{S}} \phi_{s,a}(s'')} = \frac{o_{s,a,s'} + 1}{O_{s,a} + |\mathcal{S}|},$$

and

$$\sigma^2_{s,a}(s') = \mathbb{E}_{p\sim\phi_{s,a}}[(p(s') - \bar{p}_{s,a}(s'))^2] = \frac{\phi_{s,a}(s')(\|\phi_{s,a}\|_1 - \phi_{s,a}(s'))}{\|\phi_{s,a}\|_1^2(\|\phi_{s,a}\|_1 + 1)} \leq \frac{1}{\|\phi_{s,a}\|_1} = \frac{1}{O_{s,a} + |\mathcal{S}|}.$$

Notice

$$
\begin{aligned}
|p(s') - \hat{p}_{s,a}(s')| &\leq |p(s') - \bar{p}_{s,a}(s')| + |\bar{p}_{s,a}(s') - \hat{p}_{s,a}(s')| \\
&= |p(s') - \bar{p}_{s,a}(s')| + \left| \frac{o_{s,a,s'} + 1}{O_{s,a} + |\mathcal{S}|} - \frac{o_{s,a,s'}}{O_{s,a}} \right| \\
&\leq |p(s') - \bar{p}_{s,a}(s')| + \frac{1}{O_{s,a}}.
\end{aligned}
$$

Since $|\mathcal{S}| \geq 1, \alpha \leq 1, O_{s,a} \geq 1, \sqrt{\frac{|\mathcal{S}|}{\alpha}} O_{s,a}^{-\frac{1}{3}} - \frac{1}{O_{s,a}} \geq 0$. Then, with probability at least $1 - \frac{\alpha}{|\mathcal{S}|} O_{s,a}^{-\frac{1}{3}}$,

$$\mathbb{P}_{p \sim \phi_{s,a}} \left( |p(s') - p_{s,a}^c(s')| \geq 3\sqrt{\frac{|\mathcal{S}|}{\alpha}} O_{s,a}^{-\frac{1}{3}} \right)$$

$$\leq \mathbb{P}_{p \sim \phi_{s,a}} \left( |p(s') - \bar{p}_{s,a}(s')| \geq 3\sqrt{\frac{|\mathcal{S}|}{\alpha}} O_{s,a}^{-\frac{1}{3}} - |\bar{p}_{s,a}(s') - \hat{p}_{s,a}(s')| - |\hat{p}_{s,a}(s') - p_{s,a}^c(s')| \right)$$

$$\leq \mathbb{P}_{p \sim \phi_{s,a}} \left( |p(s') - \bar{p}_{s,a}(s')| \geq 3\sqrt{\frac{|\mathcal{S}|}{\alpha}} O_{s,a}^{-\frac{1}{3}} - \frac{1}{O_{s,a}} - \sqrt{\frac{|\mathcal{S}|}{\alpha}} O_{s,a}^{-\frac{1}{3}} \right)$$

$$\leq \mathbb{P}_{p \sim \phi_{s,a}} \left( |p(s') - \bar{p}_{s,a}(s')| \geq \sqrt{\frac{|\mathcal{S}|}{\alpha}} O_{s,a}^{-\frac{1}{3}} \right)$$

$$\leq \mathbb{P}_{p \sim \phi_{s,a}} \left( |p(s') - \bar{p}_{s,a}(s')| \geq \sqrt{\frac{|\mathcal{S}|}{\alpha}} O_{s,a}^{-\frac{1}{3}} \right)$$

$$\leq \frac{\alpha O_{s,a}^{\frac{2}{3}}}{|\mathcal{S}|} \sigma_{s,a}^2(s') \tag{7}$$

$$\leq \frac{\alpha}{|\mathcal{S}|},$$

where the second last inequality holds by Chebyshev inequality. Replacing $a$ with $\pi(s)$, we obtain with probability at least $1 - \frac{\alpha}{|\mathcal{S}|} O_{s,\pi(s)}^{-\frac{1}{3}} \geq 1 - \frac{\alpha}{|\mathcal{S}|} \bar{O}^{-\frac{1}{3}}$,

$$\mathbb{P}_{p \sim \phi_{s,\pi(s)}} \left( |p(s') - p_{s,\pi(s)}^c(s')| \geq 3\sqrt{\frac{|\mathcal{S}|}{\alpha}} O_{s,\pi(s)}^{-\frac{1}{3}} \right) \leq \frac{\alpha}{|\mathcal{S}|}.$$

This further implies with probability at least $1 - \frac{\alpha}{|\mathcal{S}|} \bar{O}^{-\frac{1}{3}}$,

$$\mathbb{P}_{p \sim \phi_{s,\pi(s)}} \left( \|p - p_{s,\pi(s)}^c\|_\infty \geq 3\sqrt{\frac{|\mathcal{S}|}{\alpha}} O_{s,\pi(s)}^{-\frac{1}{3}} \right) \leq \sum_{s' \in \mathcal{S}} \mathbb{P}_{p \sim \phi_{s,\pi(s)}} \left( |p(s') - p_{s,\pi(s)}^c(s')| \geq 3\sqrt{\frac{|\mathcal{S}|}{\alpha}} O_{s,\pi(s)}^{-\frac{1}{3}} \right) \leq \alpha.$$

Let $U = \{p | \|p - p_{s,\pi(s)}^c\|_\infty \leq 3\sqrt{\frac{|\mathcal{S}|}{\alpha}} O_{s,\pi(s)}^{-\frac{1}{3}} \}$. Since with probability at least $1 - \frac{\alpha}{|\mathcal{S}|} \bar{O}^{-\frac{1}{3}}$, $\mathrm{VaR}_\alpha$ is the $\alpha$-quantile, and $\mathbb{P}_{\phi_{s,a}}(U) \geq 1 - \alpha$, we have

$$\inf_{p \in U} f(p|s, \pi(s), V) \leq (\mathrm{VaR}_\alpha)_{p \sim \phi_{s,\pi(s)}} (f(p|s, \pi(s), V)) \leq \sup_{p \in U} f(p|s, \pi(s), V).$$

Hence with probability at least $1 - \frac{\alpha}{|\mathcal{S}|} \bar{O}^{-\frac{1}{3}}$,

$$|(\mathrm{VaR}_\alpha)_{p \sim \phi_{s,\pi(s)}} (f(p|s, \pi(s), V)) - f(p_{s,\pi(s)}^c|s, \pi(s), V)|$$

$$\leq \sup_{p \in U} \|f(p|s, \pi(s), V) - f(\bar{p}_{s,\pi(s)}|s, \pi(s), V)\|$$

$$\leq \frac{|\mathcal{S}|\bar{R}}{1 - \gamma} \sup_{p \in U} \|p - \bar{p}_{s,\pi(s)}\|_\infty \tag{8}$$

$$\leq 3\sqrt{\frac{|\mathcal{S}|}{\alpha}} O_{s,\pi(s)}^{-\frac{1}{3}} \frac{|\mathcal{S}|\bar{R}}{1 - \gamma}$$

$$\leq 3\sqrt{\frac{|\mathcal{S}|}{\alpha}} \bar{O}^{-\frac{1}{3}} \frac{|\mathcal{S}|\bar{R}}{1 - \gamma},$$

where the second last equality is by (6). Since this holds for all $s$, then we obtain with probability $1 - \alpha \bar{O}^{-\frac{1}{3}}$,

$$|(\mathrm{VaR}_\alpha)_{p \sim \phi_{s,\pi(s)}} (f(p|s, \pi(s), V)) - f(\hat{p}_{s,\pi(s)}|s, \pi(s), V)| \leq 3\sqrt{\frac{|\mathcal{S}|}{\alpha}} \bar{O}^{-\frac{1}{3}} \frac{|\mathcal{S}|\bar{R}}{1 - \gamma}$$

holds for all $s \in \mathcal{S}$.

Finally, let $\mathcal{L}^{\phi,\pi}$ denote the Bellman operator for BRMDP with posterior $\phi$ and policy $\pi$, that is $V^{\phi,\pi} = \mathcal{L}^{\phi,\pi}V^{\phi,\pi} = (\text{VaR}_\alpha)_{p\sim\phi_{s,\pi(s)}}(f(p|s,\pi(s),V^{\phi,\pi}))$. Similar as Theorem 2.4, $\mathcal{L}^{\phi,\pi}$ is a $\gamma$-contraction mapping. We have

$$|V^{\phi,\pi}(s) - V^{c,\pi}(s)| = |\mathcal{L}^{\phi,\pi}V^{\phi,\pi}(s) - \mathcal{L}^{\phi,\pi}V^{c,\pi}(s) + \mathcal{L}^{\phi,\pi}V^{c,\pi}(s) - V^{c,\pi}(s)|$$
$$\leq \|\mathcal{L}^{\phi,\pi}V^{\phi,\pi} - \mathcal{L}^{\phi,\pi}V^{c,\pi}\|_\infty + |\mathcal{L}^{\phi,\pi}V^{c,\pi}(s) - V^{c,\pi}(s)|$$
$$\leq \gamma\|V^{\phi,\pi} - V^{c,\pi}\|_\infty + |\mathcal{L}^{\phi,\pi}V^{c,\pi}(s) - V^{c,\pi}(s)|.$$

Since $\|V^{c,\pi}\|_\infty \leq \frac{\bar{R}}{1-\lambda}$, we have the second term $|\mathcal{L}^{\phi,\pi}V^{c,\pi} - V^{c,\pi}| \leq 3\sqrt{\frac{|\mathcal{S}|}{\alpha}}\bar{O}^{-\frac{1}{3}}\frac{|\mathcal{S}|\bar{R}}{1-\gamma}$ by (8). Maximize over $s$ on both sides, we obtain

$$\|V^{\phi,\pi} - V^{c,\pi}\|_\infty \leq \lambda\|V^{\phi,\pi} - V^{c,\pi}\|_\infty + \bar{O}^{-\frac{1}{3}}\sqrt{\frac{|\mathcal{S}|}{\alpha}}\frac{3|\mathcal{S}|\bar{R}}{(1-\gamma)}.$$

Hence,

$$\|V^{\phi,\pi} - V^{c,\pi}\|_\infty \leq \bar{O}^{-\frac{1}{3}}\sqrt{\frac{|\mathcal{S}|}{\alpha}}\frac{3|\mathcal{S}|\bar{R}}{(1-\gamma)^2} \leq \bar{O}^{-\frac{1}{3}}\sqrt{\frac{|\mathcal{S}|}{\alpha}}\frac{5|\mathcal{S}|\bar{R}}{(1-\gamma)^2}.$$

For $\rho$ is $\text{CVaR}_\alpha$, notice by (7), we have with probability at least $1 - \frac{\alpha}{|\mathcal{S}|}O_{s,a}^{-\frac{1}{3}}$,

$$\mathbb{P}_{p\sim\phi_{s,a}}\left(|p(s') - p_{s,a}^c(s')| \geq 3\sqrt{\frac{|\mathcal{S}|}{\alpha}}O_{s,a}^{-\frac{1}{3}}\right) \leq \frac{\alpha}{|\mathcal{S}|}O_{s,a}^{-\frac{1}{3}}$$

Hence, with probability at least $1 - \frac{\alpha}{|\mathcal{S}|}O_{s,\pi(s)}^{-\frac{1}{3}} \geq 1 - \frac{\alpha}{|\mathcal{S}|}\bar{O}^{-\frac{1}{3}}$,

$$\mathbb{P}_{p\sim\phi_{s,\pi(s)}}\left(\|p - p_{s,\pi(s)}^c\|_\infty \geq 3\sqrt{\frac{|\mathcal{S}|}{\alpha}}O_{s,\pi(s)}^{-\frac{1}{3}}\right) \leq \alpha O_{s,\pi(s)}^{-\frac{1}{3}}.$$

Recall $\text{CVaR}_\alpha(X)$ is the conditional expectation on $\{X \leq \text{VaR}_\alpha(X)\}$. Let $U = \{p|\|p-p_{s,\pi(s)}^c\|_\infty \leq 3\sqrt{\frac{|\mathcal{S}|}{\alpha}}O_{s,\pi(s)}^{-\frac{1}{3}}\}$ and $W = \{p|f(p|s,\pi(s),V) \leq (\text{VaR}_\alpha)_{p\sim\phi_{s,\pi(s)}}(f(p|s,\pi(s),V))\}$ for $\|V\|_\infty \leq \frac{\bar{R}}{1-\lambda}$. Then, with probability at least $1 - \frac{\alpha}{|\mathcal{S}|}\bar{O}^{-\frac{1}{3}}$,

$$|(\text{CVaR}_\alpha)_{p\sim\phi_{s,\pi(s)}}(f(p|s,\pi(s),V)) - f(\hat{p}_{s,\pi(s)}|s,\pi(s),V)|$$
$$= \frac{1}{\alpha}\left|\int_{W\cap U}(f(p|s,\pi(s),V)) - f(\hat{p}_{s,\pi(s)}|s,\pi(s),V)\text{d}\mathbb{P}_{\phi_{s,\pi(s)}} + \frac{1}{\alpha}\int_{W\cap U^c}(f(p|s,\pi(s),V)) - f(\hat{p}_{s,\pi(s)}|s,\pi(s),V)\text{d}\mathbb{P}_{\phi_{s,a}}\right|$$
$$\leq \frac{1}{\alpha}\int_{W\cap U}\left|(f(p|s,\pi(s),V)) - f(\hat{p}_{s,\pi(s)}|s,\pi(s),V)\right|\text{d}\mathbb{P}_{\phi_{s,\pi(s)}} + \frac{1}{\alpha}\int_{W\cap U^c}\left|(f(p|s,\pi(s),V)) - f(\hat{p}_{s,\pi(s)}|s,\pi(s),V)\right|\text{d}\mathbb{P}_{\phi_{s,}}$$
$$\leq \frac{1}{\alpha}\left\{\alpha \cdot O_{s,\pi(s)}^{-\frac{1}{3}}\frac{2\bar{R}}{1-\gamma} + \alpha \cdot 3\sqrt{\frac{|\mathcal{S}|}{\alpha}}O_{s,\pi(s)}^{-\frac{1}{3}}\frac{|\mathcal{S}|\bar{R}}{1-\gamma}\right\}$$
$$\leq O_{s,\pi(s)}^{-\frac{1}{3}}\sqrt{\frac{|\mathcal{S}|}{\alpha}}\frac{5|\mathcal{S}|\bar{R}}{1-\gamma}$$
$$\leq \bar{O}^{-\frac{1}{3}}\sqrt{\frac{|\mathcal{S}|}{\alpha}}\frac{5|\mathcal{S}|\bar{R}}{1-\gamma}.$$

Then, following the same proof as for $\text{VaR}_\alpha$, with probability at least $1 - \alpha\bar{O}^{\frac{1}{3}}$, we can bound

$$\|V^{\phi,\pi} - V^{c,\pi}\|_\infty \leq \bar{O}^{-\frac{1}{3}}\sqrt{\frac{|\mathcal{S}|}{\alpha}}\frac{5|\mathcal{S}|\bar{R}}{(1-\gamma)^2}.$$

This completes the proof of the first part. For the second part, let $\pi^c$ be the optimal policy for $\mathcal{M}^c$. Then we have for both $\text{VaR}_\alpha$ and $\text{CVaR}_\alpha$, with probability at least $1 - \alpha\bar{O}^{-\frac{1}{3}}(\pi^c) \geq 1 - \alpha\underline{O}^{-\frac{1}{3}}$,

$$V^{\phi,*} \geq V^{\phi,\pi^c} \geq V^{c,\pi^c} - \underline{O}^{-\frac{1}{3}}\sqrt{\frac{|\mathcal{S}|}{\alpha}}\frac{5|\mathcal{S}|\bar{R}}{(1-\gamma)^2}.$$

Recall $\pi^*$ is the optimal policy for BRMDP, then

$$V^{c,\pi^c} \geq V^{c,\pi^*} \geq V^{\phi,*} - \underline{Q}^{-\frac{1}{3}}\sqrt{\frac{|\mathcal{S}|}{\alpha}}\frac{5|\mathcal{S}|\bar{R}}{(1-\gamma)^2}.$$

Combining together the proof is completed.

$\square$

**Proof of Proposition 4.2**

*Proof.* First notice

$$
\begin{aligned}
||Q^{\phi^t,*} - Q^{\omega,*}||_\infty &= ||\mathcal{T}^{\phi^t}Q^{\phi^t,*} - \mathcal{T}^{\phi^\omega}Q^{\omega,*}||_\infty \\
&= ||(\mathcal{T}^{\phi^t}Q^{\phi^t,*} - \mathcal{T}^{\phi^t}Q^{\omega,*}) + (\mathcal{T}^{\phi^t}Q^{\omega,*} - \mathcal{T}^{\phi^\omega}Q^{\omega,*})||_\infty \\
&\leq \gamma||Q^{\phi^t,*} - Q^{\omega,*}||_\infty + ||\mathcal{T}^{\phi^t}Q^{\omega,*} - \mathcal{T}^{\phi^\omega}Q^{\omega,*}||_\infty.
\end{aligned}
$$

Since $||Q^{\omega,*}||_\infty \leq \frac{\bar{R}}{1-\gamma}$ almost surely, we can prove the convergence of $Q^{\phi^t,*}$ if we can show that almost surely for each $(s,a) \in \mathcal{S} \times \mathcal{A}$,

$$\sup_{||Q||_\infty \leq \frac{\bar{R}}{1-\gamma}} |\mathcal{T}^{\phi^t}Q(s,a) - \mathcal{T}^{\phi^\omega}Q(s,a)| \to 0 \text{ as } t \to \infty. \tag{9}$$

Fix a state-action pair $(s,a)$, then for any observation process $\omega$, if $||O_{s,a}^\infty(\omega)||_\infty < \infty$, $\phi_{s,a}^t = \phi_{s,a}^\omega$ after some time stage $\tau$ and (9) clearly holds for such $\omega$. Otherwise, $||O_{s,a}^\infty(\omega)||_\infty = \infty$. For those $\omega$'s, by Bayesian consistency [6], we know that for any neighborhood of $p_{s,a}^c$ with parameter $\varepsilon > 0$, which is defined as $U_\varepsilon = \left\{ p \in \mathbb{R}_+^{|\mathcal{S}|} \mid \sum_{x \in \mathcal{S}} p(x) = 1, |p(x) - p_{s,a}^c(x)| \leq \varepsilon, \forall x \in \mathcal{S} \right\}$, $\lim_{t \to \infty} \mathbb{P}(p \in U_\varepsilon | \phi_{s,a}^t) = 1$ almost surely.

For any probability mass function $p \in \mathbb{R}_+^{|\mathcal{S}|}$, Recall $f(p|s,a,Q) = \mathbb{E}_{s' \sim p}[r(s,a,s') + \gamma \max_{b \in \mathcal{A}} Q(s',b)]$, we have $\forall p \in U_\varepsilon$,

$$
\begin{aligned}
|f(p|s,a,Q) - f(p_{s,a}^c|s,a,Q)| &= \left| \sum_{x \in \mathcal{S}} [p(x) - p_{s,a}^c(x)][r(s,a,x) + \gamma \max_{b \in \mathcal{A}} Q(x,b)] \right| \\
&\leq \varepsilon|\mathcal{S}|[\bar{R} + \gamma||Q||_\infty] \\
&\leq \varepsilon\frac{|\mathcal{S}|\bar{R}}{1-\gamma}
\end{aligned}
$$

In the following we fix a sample path $\omega$ such that $||O_{s,a}^\infty(\omega)||_\infty = \infty$ and drop the notation of $\omega$ for simplicity. We have almost surely:

- If $\rho$ is $\text{VaR}_\alpha$: Since $\mathbb{P}(p \in U_\varepsilon | \phi_{s,a}^t) \to 1$, for any risk level $\alpha > 0$, $\mathbb{P}(p \in U_\varepsilon | \phi_{s,a}^t) > 1 - \alpha$ for $t$ large enough. Since $\text{VaR}_\alpha$ is the $\alpha$-quantile, we have

$$\text{VaR}_\alpha^{\phi_{s,a}^t}(f(p|s,a,Q)) \geq \inf_{p \in U_\varepsilon} f(p|s,a,Q) \geq f(p_{s,a}^c|s,a,Q) - \varepsilon\frac{|\mathcal{S}|\bar{R}}{1-\gamma}. \tag{10}$$

  Similarly,

$$\text{VaR}_\alpha^{\phi_{s,a}^t}(f(p|s,a,Q)) \leq \sup_{p \in U_\varepsilon} f(p|s,a,Q) \leq f(p_{s,a}^c|s,a,Q) + \varepsilon\frac{|\mathcal{S}|\bar{R}}{1-\gamma}.$$

  Combining the two inequalities above, we have

$$|\text{VaR}_\alpha^{\phi_{s,a}^t}(f(p|s,a,Q)) - f(p_{s,a}^c|s,a,Q)| = |\mathcal{T}^{\phi^t}Q(s,a) - \mathcal{T}^{\phi^\omega}Q(s,a)| \leq \varepsilon\frac{|\mathcal{S}|\bar{R}}{1-\gamma} \tag{11}$$

  Since this holds for any $Q$ with $||Q||_\infty \leq \frac{\bar{R}}{1-\gamma}$ and $\varepsilon > 0$, we know

$$\lim_{t \to \infty} \sup_{||Q||_\infty < \frac{\bar{R}}{1-\gamma}} |\mathcal{T}^{\phi^t}Q(s,a) - \mathcal{T}^{\phi^\omega}Q(s,a)| = 0.$$

  This completes the proof.

- If $\rho$ is $\mathrm{CVaR}_\alpha$:

$$\mathrm{CVaR}_\alpha^{\phi_{s,a}^t}(f(p|s,a,Q)) = \mathrm{VaR}_\alpha^{\phi_{s,a}^t}(f(p|s,a,Q)) + \frac{1}{\alpha}\mathbb{E}_{q\sim\phi_{s,a}^t}[\mathrm{VaR}_\alpha^{\phi_{s,a}^t}(f(p|s,a,Q)) - f(q|s,a,Q)]^+$$

$$= \mathrm{VaR}_\alpha^{\phi_{s,a}^t}(f(p|s,a,Q)) + \frac{1}{\alpha}\mathbb{E}_{q\sim\phi_{s,a}^t,q\in U_\varepsilon}[\mathrm{VaR}_\alpha^{\phi_{s,a}^t}(f(p|s,a,Q)) - f(q|s,a,Q)]^+$$

$$+ \frac{1}{\alpha}\mathbb{E}_{q\sim\phi_{s,a}^t,q\in U_\varepsilon^c}[\mathrm{VaR}_\alpha^{\phi_{s,a}^t}(f(p|s,a,Q)) - f(q|s,a,Q)]^+$$

Since $\mathrm{VaR}_\alpha^{\phi_{s,a}^t}(f(p|s,a,Q)) \to f(p_{s,a}^c|s,a,Q)$ and $\phi_{s,a}^t(U_\varepsilon^c) \to 0$, we have for large enough $t$,

1. $|\mathrm{VaR}_\alpha^{\phi_{s,a}^t}(f(p|s,a,Q)) - f(p_{s,a}^c|s,a,Q)| \le \varepsilon$,

2. $\mathbb{E}_{q\sim\phi_{s,a}^t,q\in U_\varepsilon}[\mathrm{VaR}_\alpha^{\phi_{s,a}^t}(f(p|s,a,Q)) - f(q|s,a,Q)]^+ \le 2\varepsilon\frac{|\mathcal{S}|\bar{R}}{1-\gamma}$,

3. $\mathbb{E}_{q\sim\phi_{s,a}^t,q\in U_\varepsilon^c}[\mathrm{VaR}_\alpha^{\phi_{s,a}^t}(f(p|s,a,Q)) - f(q|s,a,Q)]^+ \le \mathbb{P}(p \in U_\varepsilon^c|\phi_{s,a}^t)\frac{2\bar{R}}{1-\gamma} \le 2\varepsilon\frac{|\mathcal{S}|\bar{R}}{1-\gamma}$.

Then we obtain

$$|\mathrm{CVaR}_\alpha^{\phi_{s,a}^t}(f(p|s,a,Q)) - f(p_{s,a}^c|s,a,Q)| \le \varepsilon\left(1 + \frac{4|\mathcal{S}|\bar{R}}{\alpha(1-\gamma)}\right)$$

Again, by arbitrary $\varepsilon$ and the uniformness in $Q$, we obtain the desired result.

$\square$

**Proof of Theorem 4.3**

*Proof.* **Proof sketch.** The complete proof is technical. We first provide a proof sketch. Intuitively, the bias term converges to $0$ since we either have posterior to concentrate on the true parameter or have an increasing sample size for the Monto Carlo estimator, both of which reduce the bias term to zero asymptotically. However, a major difficulty is uniform convergence (in terms of all possible values of $Q$ and sample size $N$), for which existing results do not give a straightforward guarantee. We prove the results by considering the two cases. First, when infinite observations are available, i.e., $||O_{s,a}^\infty|| = \infty$, we construct i.i.d. samples $h(p_i) \le f(p_i|s,a,Q)$ with the same sampled $p_i$ for an arbitrary given $\varepsilon$, where $f(p_i|s,a,Q)$ is defined as in (2). with probability at least $1 - \varepsilon$, $h(p_i)$ takes a value no less than $f(p_i|s,a,Q)$ by a constant multiple of $\varepsilon$. with probability at least $\varepsilon$, $h(p_i)$ takes a value bounded by a constant. With the help of the Stirling formula and subtle calculation, we can bound the expectation of order statistics of $h(p_i)$, which further lower bounds the Bellman operator estimator. The upper bound can be obtained in a similar way. Together we can bound the bias term. Second, in the case of limited observations, we prove a uniform convergence of empirical distribution for $f(p_i|s,a,Q)$ when $Q$ belongs to a bounded set. We carefully partition the probability space as a union of disjoint rectangular sets, to which the Glivenko-Cantelli theorem can be applied and the uniform convergence of the empirical distributions follows. Combining the two cases we complete the proof.

**Formal proof.** Again, we fix a state-action pair $(s,a)$, an observation process $\omega$ for which we drop the notation for simplicity. Let $Q$ be any Q-function such that $||Q||_\infty \le \frac{\bar{R}}{1-\gamma}$. We first prove for the VaR risk functional.

**Estimator for VaR**

we consider the two cases:

- $O_{s,a}^\infty = \infty$, then $\phi_{s,a}^t$ is consistent at the true parameter $p_{s,a}^c$. $\forall\varepsilon > 0$, $\mathbb{P}(p \in U_\varepsilon|\phi_{s,a}^t) \to 1$ almost surely for such $\omega$, where $U_\varepsilon = \left\{p \in \mathbb{R}_+^{|\mathcal{S}|}\,\big|\,\sum_{x\in\mathcal{S}}p(x) = 1, |p(x) - p_{s,a}^c(x)| \le \varepsilon, \forall x \in \mathcal{S}\right\}$. Then $\forall\varepsilon > 0$, $\mathbb{P}(p \in$

$U_\varepsilon|\phi^t_{s,a}) \geq 1 - \varepsilon$ for all large $t$. At iteration $k$ such that $t_k$ large enough, we obtain $p_1^{(k)}, p_2^{(k)}, \ldots, p_{N_{s,a}^{(k)}}^{(k)} \sim \phi^{t_k}_{s,a}$. For each $p_i^{(k)}$, we have $\mathbb{P}(p_i^{(k)} \in U_\varepsilon|\phi^{t_k}_{s,a}) \geq 1 - \varepsilon$. Furthermore, since $\phi^{t_k}_{s,a}$ is always smooth function, we can find $\Phi_i^{(k)} \in U_\varepsilon$ such that $\mathbb{P}(p_i^{(k)} \in \Phi_i^{(k)}|\phi^{t_k}_{s,a}) = 1 - \varepsilon$. Recall

$$f(p|s, a, Q) = \sum_{s' \in \mathcal{S}} p(s')[r(s, a, s') + \gamma \max_{b \in \mathcal{A}} Q(s, a)].$$

Define

$$h(p_i^{(k)}) = \begin{cases} \inf_{p \in U_\varepsilon} f(p|s, a, Q) & \text{if } p_i^{(k)} \in \Phi_i^{(k)} \\ \inf_p f(p|s, a, Q) & \text{if } p_i^{(k)} \notin \Phi_i^{(k)} \end{cases}$$

It is easy to see $f(p_i^{(k)}|s, a, Q) \geq h(p_i^{(k)})$. Notice the distribution of sampled $p_i^{(k)}$ only depends on $\phi^{t_k}_{s,a}, N_{s,a}^{(k)}$, which are further determined by $\omega_{t_k}$. Conditioned on $\omega_{t_k}$, $h(p_i^{(k)})$, $i = 1, 2, \ldots, N_{s,a}^{(k)}$ are i.i.d. random variables taking two values $\inf_{p \in U_\varepsilon} f(p|s, a, Q)$ and $\inf_p f(p|s, a, Q)$ with probability at least $1 - \varepsilon$ and $\varepsilon$, respectively. Notice $\inf_p f(p|s, a, Q) \geq -\frac{\bar{R}}{1-\gamma}$. And

$$|\inf_{p \in U_\varepsilon} f(p|s, a, Q) - \text{VaR}_\alpha^{\phi^{t_k}_{s,a}}(f(p|s, a, Q))|$$

$$\leq |\inf_{p \in U_\varepsilon} f(p|s, a, Q) - f(p^c_{s,a}|s, a, Q)| + |f(p^c_{s,a}|s, a, Q) - \text{VaR}_\alpha^{\phi^{t_k}_{s,a}}(f(p|s, a, Q))|.$$

By (10), we have $|\inf_{p \in U_\varepsilon} f(p|s, a, Q) - f(p^c_{s,a}|s, a, Q)| \leq \varepsilon \frac{|\mathcal{S}|||Q||_\infty}{1-\gamma}$. By (11), we have $|f(p^c_{s,a}|s, a, Q) - \text{VaR}_\alpha^{\phi^{t_k}_{s,a}}(f(p|s, a, Q))| \leq \varepsilon \frac{|\mathcal{S}|||Q||_\infty}{1-\gamma}$ for $k$ large enough. Hence, we obtain

$$|\inf_{p \in U_\varepsilon} f(p|s, a, Q) - \text{VaR}_\alpha^{\phi^{t_k}_{s,a}}(f(p|s, a, Q))| \leq 2\varepsilon \frac{|\mathcal{S}|||Q||_\infty}{1-\gamma},$$

where $||Q||_\infty$ again can be bounded by $\frac{\bar{R}}{1-\gamma}$. Denote by $X_i = f(p_i^{(k)}|s, a, Q)$ and $Y_i = h(p_i^{(k)})$. We first bound the conditional expectation of the order statistic $Y_{\lceil N_{s,a}^{(k)}\alpha \rceil:N_{s,a}^{(k)}}$, i.e., $E[Y_{\lceil N_{s,a}^{(k)}\alpha \rceil:N_{s,a}^{(k)}}|\omega_{t_k}]$.

Since $\{Y_i|\omega_{t_k}\}$ are i.i.d., we can compute the conditional mass function of $Y_{\lceil n\alpha \rceil:n}$ as

$$\mathbb{P}(Y_{\lceil n\alpha \rceil:n} = \inf_p f(p|s, a, Q)|\omega_{t_k}) = \sum_{j \geq \lceil n\alpha \rceil} \binom{n}{j} \varepsilon^j (1-\varepsilon)^{n-j}. \tag{12}$$

We now lower bound the conditional expectation of $Y_{\lceil n\alpha \rceil:n}$ by upper bound (12). First, we can show $\binom{n}{j} \varepsilon^j (1-\varepsilon)^{n-j}$ is decreasing in term of $j$ for $j > n\varepsilon$. To see this, take logarithm on the right hand side we have

$$\eta(j) = \log(n!) - \log((n-j)!j!) + j \log \varepsilon + (n-j) \log(1-\varepsilon).$$

Compute the difference

$$\eta(j+1) - \eta(j) = \log(\frac{n-j}{j+1} \cdot \frac{\varepsilon}{1-\varepsilon}).$$

Then $\eta(j+1) - \eta(j) < 0 \iff j \geq (n+1)\varepsilon - 1$. Hence for $\varepsilon < \alpha$ and $j \geq n\alpha \geq n\varepsilon > (n+1)\varepsilon - 1$, $\eta(j)$ is decreasing. Hence we can upper bound (12) by

$$(n+1-\lceil n\alpha\rceil)\binom{n}{\lceil n\alpha\rceil}\varepsilon^{\lceil n\alpha\rceil}(1-\varepsilon)^{n-\lceil n\alpha\rceil}.$$

$$=(n+1-\lceil n\alpha\rceil)\frac{n!}{(n-\lceil n\alpha\rceil)!\lceil n\alpha\rceil!}\varepsilon^{\lceil n\alpha\rceil}(1-\varepsilon)^{n-\lceil n\alpha\rceil} \tag{13}$$

By Stirling Formula,

$$\sqrt{2\pi n}\left(\frac{n}{e}\right)^n < n! < \sqrt{2\pi n}\left(\frac{n}{e}\right)^n e^{\frac{1}{12n}} < 2\sqrt{2\pi n}\left(\frac{n}{e}\right)^n.$$

Then we have

$$(13) < (n+1-\lceil n\alpha\rceil)\frac{2\sqrt{2\pi n}\left(\frac{n}{e}\right)^n \varepsilon^{\lceil n\alpha\rceil}(1-\varepsilon)^{n-\lceil n\alpha\rceil}}{\sqrt{2\pi(n-\lceil n\alpha\rceil)}\left(\frac{n-\lceil n\alpha\rceil}{e}\right)^{n-\lceil n\alpha\rceil}\sqrt{2\pi\lceil n\alpha\rceil}\left(\frac{\lceil n\alpha\rceil}{e}\right)^{\lceil n\alpha\rceil}}$$

$$= (n+1-\lceil n\alpha\rceil)\frac{\sqrt{2}\varepsilon^{\lceil n\alpha\rceil}(1-\varepsilon)^{n-\lceil n\alpha\rceil}}{\sqrt{(1-\frac{\lceil n\alpha\rceil}{n})}\left(1-\frac{\lceil n\alpha\rceil}{n}\right)^{n-\lceil n\alpha\rceil}\sqrt{\pi\lceil n\alpha\rceil}\left(\frac{\lceil n\alpha\rceil}{n}\right)^{\lceil n\alpha\rceil}}$$

For $n \geq \frac{2}{\alpha(1-\alpha)}$, we have

1. $(n+1-\lceil n\alpha\rceil) \leq 2n(1-\alpha)$,
2. $\varepsilon^{\lceil n\alpha\rceil} \leq \varepsilon^{n\alpha}$,
3. $(1-\varepsilon)^{n-\lceil n\alpha\rceil} \leq (1-\varepsilon)^{n(1-\alpha)}/(1-\varepsilon)$,
4. $(\frac{\lceil n\alpha\rceil}{n})^{\lceil n\alpha\rceil} \geq (\frac{\lceil n\alpha\rceil}{n})^{n\alpha+1} \geq \alpha^{n\alpha}\cdot\alpha$,
5. $1-\frac{\lceil n\alpha\rceil}{n} \geq \frac{1-\alpha}{2}$,
6. $(1-\frac{\lceil n\alpha\rceil}{n})^{n-\lceil n\alpha\rceil} \geq (1-\frac{\lceil n\alpha\rceil}{n})^{n(1-\alpha)} \geq \left(\frac{1-\alpha}{2}\right)^{n(1-\alpha)}$.

Then

$$(13) \leq 2\sqrt{\frac{2}{\pi}}n(1-\alpha)\frac{\varepsilon^{n\alpha}(1-\varepsilon)^{n(1-\alpha)}/(1-\varepsilon)}{\sqrt{\frac{1-\alpha}{2}}(\frac{1-\alpha}{2})^{n(1-\alpha)}\sqrt{n\alpha}\alpha^{n\alpha}\cdot\alpha}$$

$$= \frac{4\sqrt{(1-\alpha)}}{(1-\varepsilon)\alpha^{\frac{3}{2}}\sqrt{\pi}}\sqrt{n}\left(\frac{\varepsilon^\alpha(1-\varepsilon)^{1-\alpha}}{\alpha^\alpha(\frac{1-\alpha}{2})^{1-\alpha}}\right)^n$$

$$= \frac{C}{1-\varepsilon}\sqrt{n}\beta^n,$$

where $C = \frac{4\sqrt{(1-\alpha)}}{\alpha^{\frac{3}{2}}\sqrt{\pi}}$ is a constant and $\beta = \frac{\varepsilon^\alpha(1-\varepsilon)^{1-\alpha}}{\alpha^\alpha(\frac{1-\alpha}{2})^{1-\alpha}}$. For $\varepsilon$ small enough, we can ensure $\beta < 1$. Let

$$\zeta(x) = \log(\sqrt{x}\beta^x) = \frac{1}{2}\log x + x\log\beta.$$

Then, since $\log\beta < 0$, $\zeta(x)$ attains the maximum at $x = -\frac{2}{\log\beta}$. Then we have

$$\frac{C}{1-\varepsilon}\sqrt{n}\beta^n = \frac{C}{1-\varepsilon}e^{\zeta(n)}$$

$$\leq \frac{C}{1-\varepsilon}e^{\zeta(-\frac{2}{\log\beta})}$$

$$= \frac{C}{1-\varepsilon}\sqrt{-\frac{2}{\log\beta}}e^{-2}$$

For $n \leq \lfloor \frac{2}{\alpha(1-\alpha)} \rfloor$, we have

$$\mathbb{P}(Y_{\lceil n\alpha \rceil:n} = \inf_p f(p|s,a,Q)|\omega_{t_k})$$

$$\leq \mathbb{P}(Y_{1:n} = \inf_p f(p|s,a,Q)|\omega_{t_k})$$

$$= 1 - (1-\varepsilon)^n$$

$$\leq 1 - (1-\varepsilon)^{\lfloor \frac{2}{\alpha(1-\alpha)} \rfloor}$$

Let

$$C'(\varepsilon) := \max\{\frac{C}{1-\varepsilon}\sqrt{-\frac{2}{\log\beta}}\mathrm{e}^{-2}, 1 - (1-\varepsilon)^{\lfloor \frac{2}{\alpha(1-\alpha)} \rfloor}\},$$

where $C'(\varepsilon) \to 0$ as $\varepsilon \to 0$.
Then we have

$$\mathbb{E}[Y_{\lceil n\alpha \rceil:n}|\omega_{t_k}]$$
$$= \mathbb{P}(Y_{\lceil n\alpha \rceil:n} = \inf_{pi} f(p|s,a,Q)|\omega_{t_k}) * (\inf_p f(p|s,a,Q)) + \mathbb{P}(Y_{\lceil n\alpha \rceil:n} = \inf_{p\in U_\varepsilon} f(p|s,a,Q)|\omega_{t_k}) * \inf_{p\in U_\varepsilon} f(p|s,a,Q)$$

$$\geq \mathbb{P}(Y_{\lceil n\alpha \rceil:n} = \inf_p f(p|s,a,Q)|\omega_{t_k}) * (-\frac{\bar{R}}{1-\gamma})$$

$$+ \mathbb{P}(Y_{\lceil n\alpha \rceil:n} = \inf_{p\in U_\varepsilon} f(p|s,a,Q)|\omega_{t_k}) * (\mathrm{VaR}_\alpha^{\phi_{s,a}^{t_k}}(f(p|s,a,Q)) - 2\varepsilon|\mathcal{S}|||Q||_\infty)$$

$$\geq C'(\varepsilon) * (-\frac{\bar{R}}{1-\gamma}) + (1 - C'(\varepsilon)) * (\mathrm{VaR}_\alpha^{\phi_{s,a}^{t_k}}(f(p|s,a,Q)) - 2\varepsilon|\mathcal{S}|\frac{\bar{R}}{1-\gamma})$$

Hence we have almost surely,

$$\mathbb{E}[X_{\lceil N_{s,a}^{(k)}\alpha \rceil:N_{s,a}^{(k)}} - \mathrm{VaR}_\alpha^{\phi_{s,a}^t}(f(p|s,a,Q))|\omega_{t_k}]$$

$$\geq \mathbb{E}[Y_{\lceil N_{s,a}^{(k)}\alpha \rceil:N_{s,a}^{(k)}} - \mathrm{VaR}_\alpha^{\phi_{s,a}^{t_k}}(f(p|s,a,Q))|\omega_{t_k}]$$

$$\geq -C'(\varepsilon)\left((\frac{\bar{R}}{1-\gamma}) + \mathrm{VaR}_\alpha^{\phi_{s,a}^{t_k}}(f(p|s,a,Q))\right) - 2\varepsilon(1-C'(\varepsilon))|\mathcal{S}|\frac{\bar{R}}{1-\gamma}$$

$$\geq -2(C'(\varepsilon) + \varepsilon(1-C'(\varepsilon))|\mathcal{S}|)\left(\frac{\bar{R}}{1-\gamma}\right)$$

$$=: -\tilde{C}(\varepsilon),$$

where $\tilde{C}(\varepsilon) \to 0$ as $\varepsilon \to 0$.
Similarly by constructing

$$h(p_i^{(k)}) = \begin{cases} \sup_{p\in U_\varepsilon} f(p|s,a,Q) & \text{if } p_i^{(k)} \in \Psi_i \\ \sup_p f(p|s,a,Q) & \text{if } p_i^{(k)} \notin \Psi_i \end{cases}$$

We can obtain

$$\mathbb{E}[X_{\lceil N_{s,a}^{(k)}\alpha \rceil:N_{s,a}^{(k)}} - \mathrm{VaR}_\alpha^{\phi_{s,a}^{t_k}}(f(p|s,a,Q))|\omega_{t_k}] \leq \widehat{C}(\varepsilon),$$

almost surely for $k$ large enough and $\widehat{C}(\varepsilon) \to 0$ as $\varepsilon \to 0$.

Notice $\mathcal{T}^{\phi^{t_k}}Q(s,a) = \mathrm{VaR}_\alpha^{\phi_{s,a}^{t_k}}(f(p|s,a,Q))$. Furthermore, since both $\tilde{C}(\varepsilon)$ and $\widehat{C}(\varepsilon)$ do not depend on $Q$ and by arbitrary $\varepsilon$, we obtain

$$\lim_{k\to\infty} \sup_{||Q||_\infty \leq \frac{\bar{R}}{1-\gamma}} \left|\mathbb{E}[\widehat{\mathcal{T}}^{(k)}Q(s,a) - \mathcal{T}^{\phi^{t_k}}Q(s,a)|\omega_{t_k}]\right| = 0.$$

- If $O_{s,a}^\infty < \infty$, then the posterior distribution on $(s,a)$ remains the same as $\phi_{s,a}^\omega$ for all large $t$ and the sample size $N_{s,a}^{(k)}$ tends to infinity. In this case, by Glivenko-Cantelli theorem,

we know almost surely the empirical distribution conditioned on $\phi_{s,a}^\omega$ (which is further determined by $\omega$), $F_{N_{s,a}^{(k)}}^{(k)}$, for the sampled $p_1^{(k)}, p_2^{(k)}, \ldots, p_{N_{s,a}^{(k)}}^{(k)}$ uniformly converges to the distribution $\phi_{s,a}^\omega$ as sample size $N_{s,a}^{(k)}$ goes to infinity. Denote by $X_i^{(k)} = f(p_i^{(k)}|s, a, Q) = \sum_{s' \in S} p_i^{(k)}(s')(r(s, a, s') + \gamma \max_{b \in \mathcal{A}} Q(s', b)) = \sum_{s' \in S} d_{s'} p_i^{(k)}(s') = d^\top p_i^{(k)}$. We want to show uniform convergence of empirical distribution of $\{X_i^{(k)}\}_{i=1,2,\ldots,N_{s,a}^{(k)}}$ conditioned on $\phi_{s,a}^\omega$ for $d \in D$ uniformly, where $D$ contains all the possible value $d$ can take. Denote by $G_n^d$ the empirical distribution conditioned on $\omega_{t_k}$ of $\{X_i^{(k)}\}_{i=1,2,\ldots,n}$ in terms of $d$ and $G^d$ the true distribution of $d^\top p$ for $p \sim \phi_{s,a}^\omega$. We want to show

$$\lim_{n \to \infty} \sup_{d \in D, x} |G_n^d(x) - G^d(x)| \to 0 \qquad \text{almost surely.}$$

Notice that $|d| \leq \frac{\bar{R}}{1-\gamma}$ and $(d + \varepsilon 1)^\top \phi = d^\top \phi + \varepsilon$, we have $G_n^d(x) = G_n^{d+\varepsilon 1}(x + \varepsilon)$. Hence we only need to prove for $D = \{d \in R^{|\mathcal{S}|} : M\mathbf{1} \geq d \geq \mathbf{1}\}$, where $M = \frac{2\bar{R}}{1-\gamma} + 1$. Denote by $\mathbb{P}_n(K) = \frac{1}{n} \sum_{i=1}^n \mathbf{1}\{p_i \in K\}$ where $p_1, \cdots, p_n \sim \phi_{s,a}^\omega$ are i.i.d samples.

For an arbitrary $\varepsilon = \frac{1}{\tilde{N}}$, we have

$$\mathbb{P}_n\left(\sum_{s=1}^{|\mathcal{S}|} d_s q_s \leq x\right)$$

$$= \sum_{k_2=1}^{\tilde{N}} \cdots \sum_{k_{|\mathcal{S}|}=1}^{\tilde{N}} \mathbb{P}_n(\sum_{s=1}^{|\mathcal{S}|} d_s q_s \leq x, (k_i - 1)\varepsilon < q_i \leq k_i \varepsilon, i = 2, \ldots, |\mathcal{S}|)$$

$$\leq \sum_{k_2=1}^{\tilde{N}} \cdots \sum_{k_{|\mathcal{S}|}=1}^{\tilde{N}} \mathbb{P}_n\left(q_1 \leq \frac{1}{d_1}(x - \sum_{s=2}^{|\mathcal{S}|} d_s(k_s - 1)\varepsilon), (k_i - 1)\varepsilon < q_i \leq k_i \varepsilon, i = 2, \ldots, |\mathcal{S}|\right)$$

Since the last probability is just some probability of rectangulars, it can be expressed by finite number of cumulative distribution function, which by the Glivenko-Cantelli theorem, uniformly converge to the true distribution. Then, almosdt surely there exists $n_\varepsilon > 0$ (independent of $d, \varepsilon, k_i$), for $n \geq n_\varepsilon$, we have each

$$\mathbb{P}_n\left(q_1 \leq \frac{1}{d_1}(x - \sum_{s=2}^{|\mathcal{S}|} d_s(k_s - 1)\varepsilon), (k_i - 1)\varepsilon < q_i \leq k_i \varepsilon, i = 2, \ldots, |\mathcal{S}|\right)$$

$$\leq \mathbb{P}_{q \sim \phi_{s,a}^\omega}\left(q_1 \leq \frac{1}{d_1}(x - \sum_{s=2}^{|\mathcal{S}|} d_s(k_s - 1)\varepsilon), (k_i - 1)\varepsilon < q_i \leq k_i \varepsilon, i = 2, \ldots, |\mathcal{S}|\right) + \frac{1}{\tilde{N}^{|\mathcal{S}|}}$$

Hence we obtain

$$\mathbb{P}_n\left(\sum_{s=1}^{|\mathcal{S}|} d_s q_s \leq x\right)$$

$$\leq \sum_{k_2=1}^{\tilde{N}} \cdots \sum_{k_{|\mathcal{S}|}=1}^{\tilde{N}} \mathbb{P}_{q \sim \phi_{s,a}^\omega}\left(q_1 \leq \frac{1}{d_1}(x - \sum_{s=2}^{|\mathcal{S}|} d_s(k_s - 1)\varepsilon), (k_i - 1)\varepsilon < q_i \leq k_i \varepsilon, i = 2, \ldots, |\mathcal{S}|\right) + \frac{1}{\tilde{N}}$$

$$\leq \sum_{k_2=1}^{\tilde{N}} \cdots \sum_{k_{|\mathcal{S}|}=1}^{\tilde{N}} \mathbb{P}_{q \sim \phi_{s,a}^\omega}\left(\sum_{s=1}^{|\mathcal{S}|} d_s q_s \leq x + \sum_{s=2}^{|\mathcal{S}|} d_s \varepsilon, (k_i - 1)\varepsilon < q_i \leq k_i \varepsilon, i = 2, \ldots, |\mathcal{S}|\right) + \frac{1}{\tilde{N}}$$

$$= \mathbb{P}_{q \sim \phi_{s,a}^\omega}\left(\sum_{s=1}^{|\mathcal{S}|} d_s q_s \leq x + \sum_{s=2}^{|\mathcal{S}|} d_s \varepsilon\right) + \frac{1}{\tilde{N}}$$

$$\leq \mathbb{P}_{q \sim \phi_{s,a}^\omega}\left(\sum_{s=1}^{|\mathcal{S}|} d_s q_s \leq x + \frac{(|\mathcal{S}| - 1)M}{\tilde{N}}\right) + \frac{1}{\tilde{N}}$$

Since this holds for any $d, x$, we have

$$\sup_{d \in D, x \in R} \left\{ \mathbb{P}_n \left( \sum_{s=1}^{|\mathcal{S}|} d_s q_s \leq x \right) - \mathbb{P}_{q \sim \phi_{s,a}^\omega} \left( \sum_{s=1}^{|\mathcal{S}|} d_s q_s \leq x \right) \right\}$$

$$\leq \frac{1}{\widetilde{N}} + \sup_{d \in D, x \in R} \left\{ \mathbb{P}_{q \sim \phi_{s,a}^\omega} \left( \sum_{s=1}^{|\mathcal{S}|} d_s q_s \leq x + \frac{(|\mathcal{S}| - 1)M}{\widetilde{N}} \right) - \mathbb{P}_{q \sim \phi_{s,a}^\omega} \left( \sum_{s=1}^{|\mathcal{S}|} d_s q_s \leq x \right) \right\}$$

The last term converges to zero as $\widetilde{N} \to \infty$. This is indicated by

$$\mathbb{P}_{q \sim \phi_{s,a}^\omega} \left( \sum_{s=1}^{|\mathcal{S}|} d_s q_s \leq x + \frac{(|\mathcal{S}| - 1)M}{\widetilde{N}} \right) - \mathbb{P}_{q \sim \phi_{s,a}^\omega} \left( \sum_{s=1}^{|\mathcal{S}|} d_s q_s \leq x \right)$$

$$= \mathbb{P}_{q \sim \phi_{s,a}^\omega} \left( x < \sum_{s=1}^{|\mathcal{S}|} d_s q_s \leq x + \frac{(|\mathcal{S}| - 1)M}{\widetilde{N}} \right)$$

$$\leq \sum_{s=1}^{\mathcal{S}} \mathbb{P}_{q \sim \phi_{s,a}^\omega} \left( \frac{x}{d_s} < q_s \leq \frac{x}{d_s} + \frac{(|\mathcal{S}| - 1)M}{d_s \widetilde{N}} \right)$$

$$\leq \sum_{s=1}^{\mathcal{S}} \mathbb{P}_{q \sim \phi_{s,a}^\omega} \left( \frac{x}{d_s} < q_s \leq \frac{x}{d_s} + \frac{(|\mathcal{S}| - 1)M}{\widetilde{N}} \right)$$

Since each $0 \leq q_s \leq 1$ and its marginal distribution $\phi_{s,a}^\omega(q_s)$ is continuous, it is uniformly continuous in $0 \leq q \leq 1$. Hence each $\mathbb{P}_{q \sim \phi_{s,a}^\omega} \left( \frac{x}{d_s} < q_s \leq \frac{x}{d_s} + \frac{(|\mathcal{S}|-1)M}{\widetilde{N}} \right)$ converge uniformly to 0 as $\widetilde{N} \to \infty$. Hence by arbitrary $\widetilde{N}$, we have

$$\limsup_{n \to \infty} \sup_{d \in D, x \in R} \left\{ \mathbb{P}_n \left( \sum_{s=1}^{|\mathcal{S}|} d_s q_s \leq x \right) - \mathbb{P}_{q \sim \phi_{s,a}^\omega} \left( \sum_{s=1}^{|\mathcal{S}|} d_s q_s \leq x \right) \right\} \leq 0.$$

Similarly, we can obtain the other side as

$$\liminf_{n \to \infty} \inf_{d \in D, x \in R} \left\{ \mathbb{P}_n \left( \sum_{s=1}^{|\mathcal{S}|} d_s q_s \leq x \right) - \mathbb{P}_{q \sim \phi_{s,a}^\omega} \left( \sum_{s=1}^{|\mathcal{S}|} d_s q_s \leq x \right) \right\} \geq 0.$$

Hence,

$$\sup_{d \in D, x} |G_n^d(x) - G^d(x)| \to 0 \qquad \text{almost surely.}$$

We obtain the uniform convergence for empirical distribution in $d$, or equivalently, $Q$, conditioned on the observation process $\omega$. Hence the quantile estimator $X_{\lceil n\alpha \rceil : n} \to \text{VaR}_\alpha^{\phi_{s,a}^\omega}(Q)$ uniformly in $Q$ almost surely. Furthermore, since $X_i$ is bounded,

$$\lim_{k \to \infty} \sup_{||Q||_\infty \leq \frac{\bar{R}}{1-\gamma}} \left| \mathbb{E}[\widehat{\mathcal{T}}^{(k)} Q(s,a) - \mathcal{T}^{\phi^{t_k}} Q(s,a) | \omega_{t_k}] \right| = 0 \qquad \text{almost surely .}$$

**Estimator for CVaR**

Given sample $p_1^{(k)}, p_2^{(k)}, \ldots, p_{N_{s,a}^{(k)}}^{(k)}$ and $X_i = f(p_i^{(k)} | s, a, Q) = \sum_{s' \in \mathcal{S}} p_i^{(k)}(s')[r(s, a, s') + \gamma \max_{b \in \mathcal{A}} Q(s', b)]$. The Monto Carlo estimator for $\text{CVaR}_\alpha^{\phi_{s,a}^{t_k}}(f(p|s, a, Q))$ is $\frac{1}{\lceil N_{s,a}^{(k)} \alpha \rceil} \sum_{j=1}^{\lceil N_{s,a}^{(k)} \alpha \rceil} X_{j : N_{s,a}^{(k)}}$. Again we consider the two cases:

- $O_{s,a}^\infty = \infty$. Then we know almost surely $\phi_{s,a}^t$ is consistent at $p_{s,a}^c$. Then by the proof for bias of VaR estimator, $\forall 1 > \varepsilon > 0$, $\mathbb{E}[X_{\lceil N_{s,a}^{(k)} \varepsilon \rceil : N_{s,a}^{(k)}} | \omega] \to \text{VaR}_\varepsilon^{\phi_{s,a}^{t_k}}(f(p|s, a, Q))$ as

$k \to \infty$ uniformly in $Q$ almost surely. Then

$$\mathbb{E}\left[\frac{1}{\lceil N_{s,a}^{(k)}\alpha \rceil} \sum_{j=1}^{\lceil N_{s,a}^{(k)}\alpha \rceil} X_{j:N_{s,a}^{(k)}}|\omega_{t_k}\right]$$

$$=\mathbb{E}\left[\frac{1}{\lceil N_{s,a}^{(k)}\alpha \rceil} \sum_{j=1}^{\lceil N_{s,a}^{(k)}\varepsilon \rceil} X_{j:N_{s,a}^{(k)}}|\omega_{t_k}\right] + \mathbb{E}\left[\frac{1}{\lceil N_{s,a}^{(k)}\alpha \rceil} \sum_{j=\lceil N_{s,a}^{(k)}\varepsilon \rceil}^{\lceil N_{s,a}^{(k)}\alpha \rceil} X_{j:N_{s,a}^{(k)}}|\omega_{t_k}\right]$$

$$\geq -\frac{\lceil N_{s,a}^{(k)}\varepsilon \rceil}{\lceil N_{s,a}^{(k)}\alpha \rceil}\frac{\bar{R}}{1-\gamma} + \frac{\lceil N_{s,a}^{(k)}\alpha \rceil - \lceil N_{s,a}^{(k)}\varepsilon \rceil}{\lceil N_{s,a}^{(k)}\alpha \rceil}\mathbb{E}[X_{\lceil N_{s,a}^{(k)}\varepsilon \rceil:N_{s,a}^{(k)}}|\omega_{t_k}]$$

Since $\mathbb{E}[X_{\lceil N_{s,a}^{(k)}\varepsilon \rceil:N_{s,a}^{(k)}}|\phi_{s,a}^{t_k}] \to \mathrm{VaR}_\varepsilon^{\phi_{s,a}^{t_k}}(f(p|s,a,Q))$, $\mathrm{VaR}_\varepsilon^{\phi_{s,a}^{t_k}}(f(p|s,a,Q)) \to$ $f(p_{s,a}^c|s,a,Q)$, $\mathrm{CVaR}_\alpha^{\phi_{s,a}^{t}}(f(p|s,a,Q)) \to f(p_{s,a}^c|s,a,Q)$ uniformly for $-\frac{\bar{R}}{1-\gamma} \leq Q \leq \frac{\bar{R}}{1-\gamma}$ as $k \to \infty$ and $\varepsilon$ is chosen arbitrarily, we have $\liminf_{k\to\infty}\{\mathbb{E}\left[\frac{1}{\lceil N_{s,a}^{(k)}\alpha \rceil}\sum_{j=1}^{\lceil N_{s,a}^{(k)}\alpha \rceil} X_{j:N_{s,a}^{(k)}}|\omega_{t_k}\right] - \mathrm{CVaR}_\alpha^{\phi_{s,a}^{t_k}}(f(p|s,a,Q))\} \geq$ 0 uniformly in $Q$ almost surely. Similarly we can also prove $\limsup_{k\to\infty}\{\mathbb{E}\left[\frac{1}{\lceil n\alpha \rceil}\sum_{j=1}^{\lceil n\alpha \rceil} X_{j:n}|\omega_{t_k}\right] - \mathrm{CVaR}_\alpha^{\phi_{s,a}^{t_k}}(f(p|s,a,Q))\} \leq 0$ uniformly in $Q$ almost surely. Combining the two inequalities together, we obtain

$$\lim_{k\to\infty} \sup_{||Q||_\infty \leq \frac{\bar{R}}{1-\gamma}} \left|\mathbb{E}[\widehat{\mathcal{T}}^{(k)}Q(s,a) - \mathcal{T}^{\phi^{t_k}}Q(s,a)|\omega_{t_k}]\right| = 0 \qquad \text{almost surely}.$$

- For $O_{s,a}^\infty(\infty) < \infty$. By proof of VaR estimator we know the empirical distribution of $f(p_i^{(k)}|s,a,Q)$ conditioned on $\omega$, converge uniformly to $\phi_{s,a}^\omega$. Hence the CVaR estimator also has the desired result.

$\square$

**Lemma B.3.** *[24] Consider a stochastic process $(\alpha_t, \Delta_t, g)$, $t \geq 0$, where $\alpha_t, \Delta_t, g : X \to \mathfrak{R}$ satisfy the equations*

$$\Delta_{t+1}(x) = (1 - \alpha_t(x))\Delta_t(x) + \alpha_t(x)g(x), \quad x \in X, \quad t = 0,1,2,\dots$$

*Let $P_t$ be a sequence of increasing $\sigma$-fields such that $\alpha_0$ and $\Delta_0$ are $P_0$-measurable and $\alpha_t, \Delta_t$ and $F_{t-1}$ are $P_t$-measurable. Let $||\cdot||_W$ denote some weighted maximum norm. Assume that the following hold:*

1. *the set $X$ is finite.*

2. *$0 \leq \alpha_t(x) \leq 1, \sum_t \alpha_t(x) = \infty, \sum_t \alpha_t^2(x) < \infty$ w.p.1.*

3. *$||E\{g(\cdot) \mid P_t\}||_W \leq \kappa ||\Delta_t||_W + c_t$, where $\kappa \in [0,1)$ and $c_t$ converges to zero w.p.1.*

4. *$\mathrm{Var}\{g(x) \mid P_t\} \leq K(1 + ||\Delta_t||_W)^2$, where $K$ is some constant.*

*Then, $\Delta_t$ converges to zero with probability one (w.p.1).*

**Proof for Theorem 4.4.**

*Proof.* Following the same notation in Theorem 4.3. Denote by $Q^{(k)}$ the Q-function in iteration $k$. Define

$$\mathrm{bias}_{(k)}(s,a) = \mathbb{E}[\widehat{\mathcal{T}}^{(k)}Q^{(k)}(s,a) - \mathcal{T}^{\phi^{t_k}}Q^{(k)}(s,a)|\omega_{t_k}].$$

Then by Theorem 4.3 and observing $||Q^{(k)}|| < \frac{\bar{R}}{1-\gamma}$ almost surely, we have almost surely,

$$\lim_{k\to\infty} \mathrm{bias}_{(k)}(s,a) = 0 \qquad \forall s \in \mathcal{S}, a \in \mathcal{A}.$$

Denote by $\Delta^{(k)} = Q^{(k)} - Q^{\omega,*}$. $\Delta^{(k)}$ satisfies

$$\Delta^{(k+1)}(s,a) = (1-\lambda_k)\Delta^{(k)} + \lambda_k F_{(k)},$$

where $F_{(k)} = \widehat{\mathcal{T}}^{(k)}Q^{(k)} - Q^{\omega,*}$. Let $\mathcal{H}_{(k)}$ be the $\sigma$-field generated by $\{\{\Delta_{(\kappa)}\}_{\kappa=1}^{k}, \{\lambda_{(\kappa)}\}_{\kappa=1}^{k}, \{F_{(\kappa)}\}_{\kappa=1}^{k-1}\}$. We have

$|\mathbb{E}[F_{(k)}(s,a)|\mathcal{H}_{(k)}]|$

$=|\mathbb{E}[\mathbb{E}[F_{(k)}(s,a)|\omega_{t_k},\mathcal{H}_{(k)}]|\mathcal{H}_{(k)}]|$

$=|\mathbb{E}[\mathcal{T}^{(k)}(Q^{(k)})(s,a) - Q^{\phi^{t_k},*}(s,a) + Q^{\phi^{t_k},*}(s,a) - Q^{\omega,*}(s,a) + \mathrm{bias}_{(k)}(s,a)|\mathcal{H}_{(k)}]|$

$=|\mathbb{E}[\widehat{\mathcal{T}}^{(k)}(Q^{(k)} - Q^{\phi^{t_k},*})|\mathcal{H}_{(k)}] + \mathbb{E}[Q^{\phi^{t_k},*}(s,a) - Q^{\omega,*}(s,a)|\mathcal{H}_{(k)}] + \mathbb{E}[\mathrm{bias}_{(k)}(s,a)|\mathcal{H}_{(k)}]|$

$\leq\gamma\mathbb{E}[\max_{s'\in\mathcal{S}}|\max_{b\in\mathcal{A}}Q^{(k)}(s',b) - \max_{b\in\mathcal{A}}Q^{\phi^{t_k},*}(s',b)||\mathcal{H}_{(k)}] + \mathbb{E}[||Q^{\phi^{t_k},*} - Q^{\omega,*}||_\infty|\mathcal{H}_{(k)}] + \mathbb{E}[|\mathrm{bias}_{(k)}(s,a)||\mathcal{H}_{(k)}]$

$\leq\gamma\mathbb{E}[||Q^{(k)} - Q^{\phi^{t_k},*}||_\infty|\mathcal{H}_{(k)}] + \mathbb{E}[||Q^{\phi^{t_k},*} - Q^{\omega,*}||_\infty|\mathcal{H}_{(k)}] + \mathbb{E}[|\mathrm{bias}_{(k)}(s,a)||\mathcal{H}_{(k)}]$

$\leq\gamma\mathbb{E}[||Q^{(k)} - Q^{\omega,*}||_\infty|\mathcal{H}_{(k)}] + (1+\gamma)\mathbb{E}[||Q^{\phi^{t_k},*} - Q^{\omega,*}||_\infty|\mathcal{H}_{(k)}] + \mathbb{E}[|\mathrm{bias}_{(k)}(s,a)||\mathcal{H}_{(k)}]$

$\leq\gamma||\Delta^{(k)}||_\infty + (1+\gamma)\mathbb{E}[||Q^{\phi^{t_k},*} - Q^{\omega,*}||_\infty|\mathcal{H}_{(k)}] + \mathbb{E}[|\mathrm{bias}_{(k)}(s,a)||\mathcal{H}_{(k)}].$

where $||\cdot||_\infty$ denote the entry-wise sup norm. The second equality holds since the Monto Carlo sampling procedure in iteration $k$ only depends on the posterior $\phi^{t_k}$ and sample size $N_{s,a}^{(k)}$, which are completely determined by the past observations $\omega_{t_k}$. By Proposition 4.2 and Theorem 4.3, we know both $||Q^{\phi^{t_k},*} - Q^{\omega,*}||_\infty$ and $|\mathrm{bias}_{(k)}(s,a)|$ converge to 0 almost surely. Since they are both bounded, by the bounded convergence theorem the last two terms in the last inequalities converge to zero almost surely. Furthermore, since $|F_{(k)}(s,a)| \leq \bar{R} + ||\Delta^{(k)}||_\infty$, we have

$$\mathrm{Var}(F_{(k)}(s,a)|\mathcal{H}_{(k)}) \leq (\bar{R} + ||\Delta^{(k)}||_\infty)^2$$
$$\leq (\bar{R}^2 + 1)(1 + ||\Delta^{(k)}||_\infty)^2.$$

By Lemma B.3, we obtain $||\Delta^{(k)}||_\infty$ converges to 0 almost surely, which completes the proof.

$\square$

