# OpenReview forum: "Bayesian Risk-Averse Q-Learning with Streaming Observations"
_NeurIPS.cc/2023/Conference — NeurIPS 2023 poster_

### Official Review · Reviewer_kQUg · 2023-06-13

**Soundness:** 3 good
**Presentation:** 3 good
**Contribution:** 3 good
**Rating:** 7
**Confidence:** 3

**Summary:**

The paper develops a Bayesian risk-averse Q-learning algorithm to tackle the setting of Bayesian risk MDP, which uses Bayesian posterior to estimate the transition model and impose a risk functional to account for the model uncertainty. The claim is that the proposed algorithm learns a "risk-averse yet optimal policy", which has theoretical guarantee of strong convergence.

**Strengths:**

The paper proposes an interesting formalization of an algorithm for Bayesian risks in MDP. The paper is scientifically sound and provides a few interesting results, among others with respect to the convergence (to the "optimal") in the context of infinite data. The paper explains why Monte-Carlo estimators are useful in practice and provides a few interesting theoretical properties. Numerical experiments provide illustrations of the theoretical analysis in the context of two relevant (small-scale) MDPs.

**Weaknesses:**

Even though related work is overall well-discussed, the paper could be more clear about the novelty of the different parts (e.g. Theorem 2.2 and Theorem 2.3 seem relatively generic and even though I'm not an expert in the BRMDP setting, I believe close theorems exist in the literature).

The notations and overall formalization might benefit from a few (minor) improvements to improve the readability (see additional comments and questions).

Additional comments:
- In Equation 1, $d_i$ and $p_i$ do not seem to be formally defined (even though we can guess what they refer to).
- In Equation 1, \rho is used with a subscript that depends on a sampling from a Dirichlet posterior but \rho was introduced line 114 without any subscript (the meaning of the subscript might not be fully obvious).
- line 97: $r$ is not defined (even though we can guess what they refer to)
- line 108: (s,a) is not in math mode (italic).
- line 109: "??" instead of a reference to the appendix.


**Questions:**

Main question:
- Theorems are mostly provided as fully original. What are the closest related theorems from the literature?
- The inventory management problem is considered in two settings as can be read in the supplementary material. The second one is described as "(...) the demand depends on the current inventory level s. (...) we will consider the case where observations are insufficient to estimate the transition probability for every state-action pair". What is actually meant by that? And also, why are there two settings described in the supplementary material but only one in the main paper?

Additional questions:
- The abstract mentions the following: "The proposed algorithm learns a risk-averse yet optimal policy that depends on the availability of real-world observations." This seems unclear from the abstract because a risk averse policy will in general need make a tradeoff with the best expected return. Do you mean "risk averse policy that converges to the optimal one in the context of unlimited data"?
- line 117, $\xi$ is defined as a probability distribution over $\mathbb R^n$, but is described as a "random vector taking values on $\mathbb R^n$". Can you clarify?
- Line 123: Could there be some intuitions for the different parts of assumption 2.1? In particular Line 124-128 gives lightweight information about how it is slightly similar to the notion of coherent risk measure but that notion and the difference with it are not detailed. Why is sub additivity not included, why is 2.1.3 important?

**Limitations:**

Some limitations are provided.

---

> ### Author Rebuttal · Authors · 2023-08-04
>
>   We appreciate the reviewer's comment. For the first main question,   our presented theorems can be classified into two groups. Theorem 2.2 and 2.3 characterize the property of BRMDP. The BRMDP in this paper is different from previous risk-sensitive MDP in that  risk functional is taken with respect to the posterior distribution to account for epistemic uncertainty, while most previous works of risk-sensitive MDPs impose the risk functional on the known transition probability to account for the aleatoric uncertainty. BRMDP is first proposed in [1], where they consider the finite horizon MDP whose optimal policy can be solved using dynamic programming, while we consider the infinite-horizon and discounted MDP. No existing theorems can be applied here due to the different formulations. But like many works on (robust) infinite-horizon and discounted MDP, we proved Theorem 2.2 in a standard way of showing the Bellman operator is a contraction mapping.  For Theorem 2.3, we do not see previous works have studied such "convergence" property, As we mentioned above, the previous works on risk-sensitive MDP mainly deal with aleatoric uncertainty, whose value function does not converge to that of the original MDP. The proof of Theorem 2.3 relies on the contracting property of the Bellman operator as provided in Theorem 2.2 and some statistical properties of risk measure VaR and CVaR, which are well studied in many previous works.  We agree with the reviewer that the proof of these 2 theorems can be regarded as an extension of previous work on robust MDP to BRMDP, with some effort dealing with risk functionals.
>
> The remaining theorems guarantee the convergence of the proposed algorithm, which follows the framework of stochastic approximation as many other works on Q-Learning. However, unlike previous work on either robust or non-robust Q-Learning, where they can obtain an unbiased estimator of the Bellman operator, we cannot do so as discussed in Section 3. Instead, we need to show a uniform convergence of our Monte Carlo estimator of the Bellman operator in Theorem 4.3, which is the most challenging and novel theoretical result of the paper. The proof of Theorem 4.3 is completely new and non-trivial.
>
> For the second main question, the sentence "the demand depends on the current inventory level" is misplaced and should be deleted. In both settings, the demand does not depend on the inventory level. We greatly thank the reviewer to point this out and we will correct this in the paper.
>     In addition,  the two settings refer to the result in Figure 4 and Figure 5, respectively. In Figure 4, the posterior is updated at the beginning of each stage with some newly arrived data. In Figure 5, the posterior is only estimated once at the beginning of first stage. It can be considered as pure offline Q-learning, which is the same setting as the two distributionally robust (DR) Q-Learning in the comparison baseline. Our purpose of showing result in Figure 2-4 is to illustrate the advantage of utilizing streaming real-world data, as we can reduce the epistemic uncertainty.  In Figure 5, when there is no source of streaming data, we want to illustrate that our Bayesian risk-averse (BR) policy possesses robustness like the other two DR policies. A DR policy has the best worst-case performance over a set of potential transition models. Here we list the performance (value function) of different policies under different transition models. The two DR policies are the most robust as in the worst case shown in Figure 5 (Poisson parameter equal to 2) they obtain the largest value function. Our BR policies with VaR and CVaR fall between the  risk-neutral policy and DR policies, showing the risk measure is a more flexible choice of risk attitude between the worst-case and  risk-neutral case.
>
> For the first additional question, the reviewer is correct. The risk-averse policy converges to the optimal policy as more data are available as the epistemic uncertainty is reduced to 0. We thank the reviewer to point this out and we will clarify this in the paper.
>
> For the second additional question, we thank the reviewer to point out this notation issue. We will correct this by deleting the distribution notation.
>
> For the third additional question, the only reason we replace the sub-additivity with assumption 2.1.3 is to make the assumption more general to include risk measure VaR, which is a widely used risk measure but not satisfies sub-additivity. In addition, we cannot simply delete the sub-additivity assumption without adding Assumption 2.1.3, because Assumption 2.1.3 is necessary to guarantee that the Bellman operator is a contraction mapping in the proof of Theorem 2.2.1.
>
> [1] Lin, Yifan, Yuxuan Ren, and Enlu Zhou. "Bayesian Risk Markov Decision Processes." Advances in Neural Information Processing Systems 35 (2022): 17430-17442.

---

> > ### Comment · Reviewer_kQUg · 2023-08-14
> > **Thanks for the clarfifications**
> >
> > Thanks for the clarifications, I keep my score of 7 unchanged.

---

> > > ### Author Response · Authors · 2023-08-16
> > >
> > > We thank the reviewer for the very early response as well as the previous comments that help improve the paper.

---

### Official Review · Reviewer_vogF · 2023-07-03

**Soundness:** 2 fair
**Presentation:** 3 good
**Contribution:** 2 fair
**Rating:** 6
**Confidence:** 4

**Summary:**

This paper extends previous work on Bayesian Risk-averse MDPs (BRMDPs), an informed alternative to an ambiguity set in the infinite horizon setting. In doing so, the authors first present a nested BRMDP formulation for the state value function. Then, they show that difference between optimal value functions for BRMDP and the true MDP is bounded. Afterwards, the authors presented a multi-stage risk-averse Q-learning algorithm with periodic posterior updates and a Monte Carlo estimator for the proposed risk-averse bellman operators. Simple simulations are presented to verify the better performance as well as the lesser variance of the proposed infinite horizon BRMDP formulation. It should be noted that the Bayesian posterior and Risk functionals are well-defined and that the state and action spaces discussed in the paper are both finite. The work does hold merit, and the findings could be communicated to a larger audience through a conference like NeuRIPS.

**Strengths:**

The paper is generally well-written:

1. The infinite horizon BRMDP formulation is novel and the accompanying recursive Bellman equations follow nicely.

2. Detailed proofs of the distance bound for the State function and the convergence analysis are provided.

3. Experimental results are discerned in a manner that relates directly to claims of risk aversion (smaller variances) and performance gains.


**Weaknesses:**

1. Apart from the assumption of the availability of a behavioral policy to generate real-world data, the rules for updating the sample sizes appear to be a heuristic at best. (ref. Algorithm 1)

2. The Monte Carlo estimators for the Bellman operators, in addition to generating real-world samples to update the Bayesian posterior, will make the algorithm extremely slow.

3. The state and action spaces under consideration appear to be finite, thus limiting the evaluation to more demanding experimental setups.


**Questions:**

1. In line 180, page 5, section 2.5, it is stated that a batch n(t) of observations is available. What is the assumption for the smallest batch size?

2. What is the reasoning behind using the Monte Carlo estimator?

3. Why is the benchmark Q-Learning on the true environment not provided in the experimental results?

4. In line 201, page 6, section 3.1, what does i.i.d sampling of probability distributions p_i mean?

5. What is the computational overhead of the proposed methodology, and how well will it scale?


**Limitations:**

The authors addressed one limitation of their work, i.e., that the behavior policy which generates the real-world data is assumed to be given. In terms of societal impact, there would be no potential negative impact of this work according to the considered ethical criterion.

---

> ### Author Rebuttal · Authors · 2023-08-04
>
>  We appreciate the reviewer's comment. For the first question, the proposed algorithm works for any batch size. For example, we can update the posterior once new data are available, in which case $n(t) \ge 1$. However, it is of future interest to control the number of Q-learning steps and real-data batch size to improve sample efficiency.
>
> For the second question, the Monte Carlo estimator is designed for estimating the Bellman operator, which depends on the posterior of transition model but not the real transition model. The real-world sample is only used for updating the posterior belief about the transition model to reduce the epistemic uncertainty. Once we update the posterior at the beginning of each stage, we turn to solving the BRMDP, the risk-averse problem instead of focusing on the original MDP. Our framework can be regarded as Episodic off-line RL, where within each episode (stage) we do not have sources of real data but only a fixed set of data that are used to construct the posterior.
>
> For the third question, the Q-Learning algorithm only uses real-world data. In our experiment setting, the number of real-world data $n(t) = 5$ is much smaller than the computing budget of Q-learning update $|\mathcal{S}|m(t) = 50$. As a result, the model-free Q-learning converges quite slowly as we only have very little real-world data. In fact, the reason for our proposed risk-averse formulation is to deal with the epistemic uncertainty caused by the (partially) lack of data, which can attribute to either highly cost real data or safety concerns.
>
> For the fourth question, recall $\phi_{s,a}$ denote the Dirichlet posterior distribution on the unknown transition probability $\mathcal{P}^c_{s,a} \in \mathbf{R}^{|\mathcal{S}|}$. Each $p_i$ is a $|\mathcal{S}|$-dimensional vector that represents a transition probability.
>
> For the fifth question, the computational overhead is mainly on generating samples from the posterior distribution to estimate the Bellman operator for each state-action pair. We restrict the choice of risk functional to CVaR, which is always risk-averse, to roughly compute the computational complexity. For each state-action pair, generating a $|\mathcal{S}|$-dimensional random vector takes $O(|\mathcal{S}|)$ time.  The number of samples needed to estimate the Bellman operator depends on the posterior update according to the proposed algorithm. If a certain state-action space is never observed, then estimating the Bellman operator can be very costly when the stage is large. Although in the algorithm we allow such sample size to go to infinity to prove the almost sure convergence, which requires the bias term to converge to 0. In practice we can often use a fixed sample size $\bar{N}$, which can be computed using some concentration bound for CVaR [1] to control the bias with some confidence level. Then, in each stage (inner loop in algorithm), we need to generate $m(t)\bar{N}|\mathcal{S}||\mathcal{A}|$ samples, which then takes $O(m(t)\bar{N}|\mathcal{S}|^2|\mathcal{A}|)$ time. Assume $m(t) = m$ for simplicity. Since the Q-function given by the algorithm is always bounded by $\frac{\bar{R}}{1-\gamma}$, which does not depend on $|\mathcal{S}|$, the sample size $\bar{N}$ does not depend on $|\mathcal{S}|$ by [1]. We have the computational overhead scale as $O(|\mathcal{S}|^2)$ in terms of the size of state space. When the state space is large, function approximation of the Q-function is of interest to improve the computation efficiency.
>
>
> [1] Thomas, P., Learned-Miller, E . Concentration Inequalities for Conditional Value at Risk. ICML(2019).

---

> > ### Comment · Reviewer_vogF · 2023-08-16
> >
> > I would like to thank the authors for taking the the time and providing detailed and satisfactory responses to my comments. Please try to incorporate as much as details as possible in a future revised version of the paper.

---

> > > ### Author Response · Authors · 2023-08-16
> > >
> > > We thank the reviewer for the response. We appreciate the comments and will definitely try our best to incorporate more details to make the paper more specific.

---

### Official Review · Reviewer_u4kt · 2023-07-06

**Soundness:** 3 good
**Presentation:** 2 fair
**Contribution:** 3 good
**Rating:** 6
**Confidence:** 3

**Summary:**

The paper adopts the Bayesian risk MDP (BRMDP) formulation to train a reinforcement learning agent to be robust against model uncertainty. Infinite-horizon discounted value function of BRMDP is defined by a nested formula, and properties for the value function are derived with VaR and CVaR being the risk functional. Furthermore, Q-learning algorithms for BRMDP are proposed using finite-sample estimators for the Bellman operator, and they are shown to converge to the optimal Q-function under some typical assumptions. Numerical experiments for the proposed risk-averse algorithm is shown and compared with its risk-neutral counterpart and other robust RL algorithms.

**Strengths:**

- The paper defines infinite-horizon discounted value for BRMDP, and Theorem 2.3 provides bounds on the difference between the value function of BRMDP to the value function of the true MDP. It shows that the risk-averse version will converge to the true value function as the number of observed transitions grows.

- Q-learning algorithms for BRMDP are proposed using finite-sample estimators for the Bellman operator with either VaR or CVaR as the risk functional. The Q-learning algorithms are shown to converge to the optimal risk-averse Q-function in a data-conditional sense, and this further implies its convergence to the true optimal Q-function given infinite amount of observations.

- Numerical experiments for the proposed risk-averse algorithm is shown and it outperforms two existing robust Q-learning algorithms.

**Weaknesses:**

- The BRMDP formulation is motivated to provide robust policies, but there is no discussion on what does "robustness" mean in the context of this paper. The only place some kind of robustness measure is mentioned is in the numerical experiments where the variations of performance is brought up in the discussion. But there is no proper definition for the variations and there is only very little discussion.

- Theorem 2.3 is claimed to show the trade-off between robustness and conservativeness, but it is hard to talk about robustness without any definition for robustness. In some sense, the upper bound has the risk level parameter $\alpha$ which may be viewed as a kind of robustness level, but no analysis is provided on how the risk level $\alpha$ would affect the behavior of the risk-averse policy and how $\alpha$ would improve the robustness of the performance in some sense.

**Questions:**

- Can we define the robustness using some metric like performance variations? How does the risk-averse policy compared with other policies in terms of some robustness metrics?

- It is shown in Theorem 2.2 that BRMDP has a unique optimal value function, but it's not clear wether the random data-conditional optimal Q-function in Definition 4.1 exists. Do we have existence for the data-conditional optimal Q-function following similar arguments?

**Limitations:**

limitations are adequately addressed in the paper.

---

> ### Author Rebuttal · Authors · 2023-08-04
>
> We appreciate the reviewer's comment. For the first question, we believe the reviewer's concern is whether the robustness of BRMDP can be shown in a more quantifiable and interpretable way. For example, in the literature of distributionally robust RL such as [1,2], they define the optimal robust policy as the policy that has best worst-case performance among a set of potential RL environments which belong to some ambiguity set constructed using some distributional metric. In the formulation of BRMDP, we also consider a set of potential environments, but with the ambiguity set to be the whole simplex space, that is, all the possible transition probabilities. In addition, each possible transition probability is assigned a posterior density and we do not consider the worst-case performance measure but some risk measure. The relation between distributionally robust MDP and risk measure has been studied in [1,2]. In [1], the author proved that when the ambiguity set satisfies some conditions,  minimizing the cost of distributionally robust MDP coincides with  some risk minimization optimization for some risk measure. It is later shown in [2] the equivalence between distributionally robust MDP and risk-sensitive MDP. However, the mapping between distributionally robust MDP and risk-sensitive MDP cannot be explicitly characterized, and the proposed BRMDP involves a nested formulation and risk functional over posterior distribution, all of which make it difficult to define the metric-like performance variations. Nonetheless, we believe it is helpful to characterize how the value function of BRMDP $V^{\phi,\pi}$ differentiates from the value function of the original true MDP $V^{c,\pi}$. While Theorem 2.6 serves this purpose, it is not intuitive enough. Our recent ongoing work shows, for a given deterministic policy $\pi$, number of transition observation $O_{s,a}$ for state-action pair $(s,a)$, total number of observation $O$, if $\\lim_{O \\rightarrow \\infty} \\frac{O_{s,\\pi(s)}}{O} = \\bar{o}_s,$
>
> Then
>  $V^{\phi,\pi}$ can be expressed as
> $$   V^{\phi,\pi} = V^{c,\pi} - \frac{1}{\sqrt{O}} C + o_p( \frac{1}{\sqrt{O}}), (2) $$
> where $C = (I - \gamma \mathcal{P}^c_\pi)^{-1}\gamma \bar{o} \mu^\pi$ is a constant, $ \mu^\pi(s) = \frac{-\sigma^\pi_s}{\alpha} \psi(\Psi^{-1}(\alpha)),
>  (\sigma^\pi_s)^2 = (V^{c,\pi})^\top diag(i_{s'}^\pi)(V^{c,\pi}),$
> $ i_{s'}^\pi = (\mathcal{P}^c_{s,\pi(s)}(s'))^{-1},$,
> $\psi, \Psi$ represents the pdf and cdf of standard normal distribution, and $o_p$ represents converge in probability. (2) indicates maximizing the value function of BRMDP
> is equivalent to maximizing the original value function minus some positive "bias" term depending on the risk level $\alpha$ and a problem-dependent variance term $\sigma_s^2$. Notice $ \frac{\psi(\Psi^{-1}(\alpha))}{\alpha} $ is decreasing in $\alpha$ and no larger than 1. A more risk-averse attitude $\alpha$ results in a larger bias term. Furthermore, this bias term diminishes in an order of $\frac{1}{\sqrt{O}}$, as when more data are collected, we are less pessimistic.
>
> For the second question, we thank the reviewer for pointing this issue out. The answer is yes. The data-conditional optimal Q-function is defined by each sample trajectory $\omega$ (which contains all the randomness). For each state-action pair $(s,a)$, the posterior $\phi^t_{s,a}$ will either remain unchanged after some period $\tau$ or concentrate on the true transition probability $P^c_{s,a}$ with probability 1. If $\phi^t_{s,a} = \phi^\tau_{s,a}$ for all $t>\tau$, then $\mathcal{T}^{\phi^\omega} Q(s,a) = \mathcal{T}^{\phi_\tau} Q(s,a)$; Otherwise we can define
> $$\\mathcal\{T\}^\{\\phi^\\omega\} Q(s,a) := \mathcal{T}^{c} Q(s,a)  = \mathbf{E}_\{P^c\_{s,a}\} [R(s,a,s') + \gamma \max\_b Q(s,b)].$$
>
> Then, $Q^{\omega,*}$ is the unique solution to $\mathcal{T}^{\phi^\omega} $, whose existence can be guaranteed with the same proof of Theorem 2.2. We will clarify this in the paper.
>
> [1] Bäuerle, Nicole, and Alexander Glauner. "Distributionally robust Markov decision processes and their connection to risk measures." Mathematics of Operations Research 47.3 (2022): 1757-1780.
>
> [2] Zhang, Runyu, Yang Hu, and Na Li. "Regularized Robust MDPs and Risk-Sensitive MDPs: Equivalence, Policy Gradient, and Sample Complexity." arXiv preprint arXiv:2306.11626 (2023).

---

> > ### Author Response · Authors · 2023-08-18
> > **Rebuttal reminder**
> >
> > Dear reviewer,
> >
> > We are grateful for the opportunity to address your valuable comments and concerns and will greatly appreciate your feedback in either discussions and scores. If you have further questions, please let us know so we can answer it before the end date of discussion on August 21 1pm EDT.
> >
> > Best regard,
> >
> > Authors

---

> > ### Comment · Reviewer_u4kt · 2023-08-19
> >
> > I appreciate the authors' detailed responses. I think this paper has good contents, but as mentioned in the response, it is difficult to define metric-like performance variations with the BRMDP formulation. The lack of a robustness metric leaves a disconnection between the theoretical results and numerical experiments. Thus I incline to keep my weak accept recommendation.

---

> > > ### Author Response · Authors · 2023-08-19
> > >
> > > The reviewer's concern is reasonable. We sincerely appreciate your feedback and understand it.

---

### Official Review · Reviewer_mBvj · 2023-07-29

**Soundness:** 4 excellent
**Presentation:** 4 excellent
**Contribution:** 4 excellent
**Rating:** 7
**Confidence:** 3

**Summary:**

The authors propose a novel multi-stage Bayesian risk-averse Q-learning algorithm to learn the optimal policy with streaming data. A central difference from existing methods is that they attach a risk functional on the future reward at each stage (nested), rather than just once on the total reward. To arrive at unbiased estimators for the Bellman operators, the authors resort to Monte Carlo. The convergence of the Q-learning algorithm is theoretically shown. The authors include empirical investigation of their method on two toy tasks and show that their method is more flexible than some other worst-case criteria.

**Strengths:**

The authors propose a technically rigorous addition to the the BRMDP literature. Their empirical results clearly show the utility of per-step risk functionals, and their extensive theoretical results include a number of - technically involved - contributions.
The centrepiece here is certainly the proof of Theorem 4.3, i.e. the uniform convergence of the Bellman operator estimator.
The paper is well-written and structured, and the resulting method has the potential to generate impact on the safe RL community.

**Weaknesses:**

While the paper convinces through its soundness and wealth of theoretical contributions, the empirical evaluation could feature more (and more complex) environments.

A fundamental weakness seem to be limitations in applying the authors' method to high-dimensional data streams.
For example, one weakness is that the authors do not extend their results to Q-learning with non-linear function approximation. I believe this could be done fairly easily, and would lay foundations for their method to be used on high-dimensional data streams.
Another weakness to address when moving to high-dimensional data streams seems the relative sample-inefficiency of  MC Bellman operator estimators (an empirical study of how efficient their proposed estimator is for high-dimensional data streams would be insightful).








**Questions:**

My questions concern possible extensions of this work, which I do not deem required for acceptance.

* Do you think that a more sample-efficient way (as MC is unbiased but generally not efficient) of arriving at unbiased Bellman operator estimators could be attained, e.g. using variational inference? This seems to be required for problems featuring high-dimensional data streams.

* The authors note that an extension to active sampling settings would be interesting. This would seem to require "deep exploration", necessitating a more fundamental Bayesian RL approach [1]. Do you see an easier way to make progress in this direction?

[1] Bayesian Bellman Operators, Mattie Fellows et al, NeurIPS 2021

**Limitations:**

The authors have adequately addressed the limitations of their work, although I would have liked to see a discussion of how their method might scale to high-dimensional data streams. I do not see any negative societal impact arising from this work.

---

> ### Author Rebuttal · Authors · 2023-08-04
>
> We appreciate the reviewer's careful review. For the first question, we agree that it is worth deriving some more sample-efficient the Bellman estimator and thank the reviewer for pointing this out.  One way to do this is to use some gradient-based method to update the Bellman estimator as in [1], where they have a distributionally robust Bellman operator (DRBO) and re-wrote it as a stochastic convex optimization. This enables the author to update their DRBO estimator by updating the decision variable given 1 sample of reward and state transition each time. In our BRMDP, if we restrict the risk functional to CVaR, we can re-write the Bellman estimator as
>
> $$\mathcal{T}^\phi Q(s,a) = CVaR_\alpha^{\phi_{s,a}} (f(Q|p,s,a)) = \sup_{\zeta} \\{
> \zeta - \frac{1}{\alpha}\mathbf{E}_{p\sim \phi} [(\zeta - f(Q|p,s,a) )^+]   \\}, (1) $$
>
>
> where $f(Q|p,s,a) = \sum_{s'\in\mathcal{S}} p(s')(R(s,a,s') + \max_{b\in \mathcal{A}}Q_{s',b}) $.
>
> Notice if we know the solution $\zeta^*$ to (1), then we can obtain an unbiased estimator by simply drawing one sample $p$ from $\phi_{s,a}$. The righthand side of the last equation in (1) belongs to convex optimization and can be solved using stochastic gradient descent (SGD). However, solving (1) to optimality can be sample-inefficient. To improve efficiency, we do not solve (1) to optimality but only conduct one step of SGD to update the current estimate of $\zeta^*$ and use it to further estimate the Bellman operator. Such an estimator is biased since we do not solve the problem to optimality.  Nonetheless, we can still expect the convergence of the algorithm if both the estimate of both $Q$ and $\zeta$ converge at appropriate rates (by setting proper learning rates). This is beyond the scope of this paper but is of interest to future research.
>
> Also, we agree with the author that variational inference can be used for complicated high-dimensional data streams  to improve sample efficiency in both updates of the posterior disttribution and estimation of the Bellman operator, which is also of future interest.
>
> For the second question, the reviewer is correct that the extension to active sampling settings requires dealing with "exploration" and "exploitation". Both our paper and [2]  given by the reviewer took the off-policy approach assuming the real data is given under an arbitrary behavior policy. By choosing a proper behavior policy, both [2] and our approach can be easily extended to the active sampling setting. One possible way of doing so is to adding some exploration bonus $Bonus(s,a)$ such as upper confidence bound. To be more specific, with the current estimation of Q-function, in state $s$ we can take action $a^*$ that maximizes $Q(s,a) + Bonus(s,a)$ to collect data samples for the sate-action pair $(s,a^*)$.
>
> [1] Liang, Zhipeng, et al. "Single-Trajectory Distributionally Robust Reinforcement Learning." arXiv preprint arXiv:2301.11721 (2023).
>
> [2] Bayesian Bellman Operators, Mattie Fellows et al, NeurIPS 2021

---

> > ### Comment · Reviewer_mBvj · 2023-08-19
> > **Thank you for your response, I will keep advocating for acceptance.**
> >
> > Thanks for your response. My score remains unchanged.

---

> > > ### Author Response · Authors · 2023-08-19
> > >
> > > We thank the reviewer for the reply.

---

### Decision · Program_Chairs · 2023-09-21

**Decision:**

Accept (poster)

**Comment:**

The meta-reviewer has reviewed the paper, the reviews, the responses, and agrees with the majority of the reviewers that this paper meets the NeurIPS standard.
It would be better if the authors could improve the experimental results.